# Partial Structure Discovery is Sufficient for No-regret Learning in Causal Bandits

**Muhammad Qasim Elahi**
Electrical and Computer Engineering
Purdue University
elahi0@purdue.edu

**Mahsa Ghasemi**
Electrical and Computer Engineering
Purdue University
mahsa@purdue.edu

**Murat Kocaoglu**
Electrical and Computer Engineering
Purdue University
mkocaoglu@purdue.edu

## Abstract

Causal knowledge about the relationships among decision variables and a reward variable in a bandit setting can accelerate the learning of an optimal decision. Current works often assume the causal graph is known, which may not always be available *a priori*. Motivated by this challenge, we focus on the causal bandit problem in scenarios where the underlying causal graph is unknown and may include latent confounders. While intervention on the parents of the reward node is optimal in the absence of latent confounders, this is not necessarily the case in general. Instead, one must consider a set of possibly optimal arms/interventions, each being a special subset of the ancestors of the reward node, making causal discovery beyond the parents of the reward node essential. For regret minimization, we identify that discovering the full causal structure is unnecessary; however, no existing work provides the necessary and sufficient components of the causal graph. We formally characterize the set of necessary and sufficient latent confounders one needs to detect or learn to ensure that all possibly optimal arms are identified correctly. We also propose a randomized algorithm for learning the causal graph with a limited number of samples, providing a sample complexity guarantee for any desired confidence level. In the causal bandit setup, we propose a two-stage approach. In the first stage, we learn the induced subgraph on ancestors of the reward, along with a necessary and sufficient subset of latent confounders, to construct the set of possibly optimal arms. We show that for our proposed algorithm, the number of intervention samples required to learn the set of possibly optimal arms scales polynomially with respect to the number of nodes. The second phase involves the application of a standard bandit algorithm, such as the UCB algorithm. We also establish a regret bound for our two-phase approach, which is sublinear in the number of rounds.

## 1 Introduction

Causal bandits have been a topic of interest since their inception and have been studied in various contexts [1]. The authors assumed precise knowledge of the causal graph and the impact of interventions or actions on the parents of the reward node. Subsequently, there has been a flurry of research on causal bandits [2, 3, 4]. The primary limitation of the majority of existing works on causal bandits is their assumption of full knowledge of the causal graph, which is often impractical for many

38th Conference on Neural Information Processing Systems (NeurIPS 2024).

real-world applications [1, 5, 6]. Recently, efforts have been made to overcome this limitation. In [7], the authors propose a sample efficient algorithm for cases where the causal graph can be represented as a directed tree or a causal forest and later extend the algorithm to encompass a broader class of general chordal graphs. However, the proposed algorithm is only applicable to scenarios where the Markov equivalence class (MEC) of the causal graph is known and does not have confounders. In [8], the authors propose a causal bandit algorithm that does not require any prior knowledge of the causal structure and leverages separating sets. However, their theoretical result holds only when a true separating set is known. The paper by Konobeev et al. [9] also deals with causal bandits with an unknown graph and proposes a two-phase approach. The first phase uses a randomized parent search algorithm to learn the parents of the reward node, and the second phase employs UCB to identify the optimal intervention over the parents of the reward node. However, similar to [7], they assume causal sufficiency, i.e., no latent confounders are present. In another related paper, [10], the authors initially emphasize the challenge of dealing with exponentially many arms when addressing causal bandits with an unknown graph. To tackle this issue, the authors assume that the reward is a noisy additive function of its parents. This assumption enables them to reframe the problem as an additive combinatorial linear bandit problem.

We also focus on the causal bandit setup where the causal graph is unknown, but we allow the presence of latent confounders and make no parametric assumptions. The optimal intervention in this case is not limited to parents of the reward node; instead, we have a candidate set of optimal interventions, called possibly optimal minimum intervention sets (POMISs), each being a special subset of the ancestors of the reward node [5]. Thus, learning only the parents of the reward, similar to [9], is insufficient. This implies that causal discovery beyond parents of the reward is imperative. However, for regret minimization, discovering the full causal structure is not necessary. Instead, we characterize the set of necessary and sufficient latent confounders one needs to detect/learn to ensure all the possibly optimal arms are learned correctly.

Causal discovery is a well-studied problem and can be applied to our setup [11, 12, 13]. However, the majority of the existing causal discovery algorithms rely on the availability of an infinite amount of interventional data [14, 15, 16]. Some prior work shows that discovery is possible with limited interventional data, with theoretical guarantees when the underlying causal graph is a tree and contains no latent confounders [17]. Also, the paper [18] proposes a sample-efficient active learning algorithm for causal graphs without latent confounders, given that the MEC for the underlying causal graph is known. Bayesian causal discovery can also be a valuable tool when interventional data is limited. However, it faces challenges when tasked with computing posterior probabilities across the combinatorial space of directed acyclic graphs (DAGs) without specific parametric assumptions [19, 20, 21]. All in all, the sample-efficient learning of causal graphs with latent confounders, without any parametric or graphical assumptions, with theoretical guarantees, remains an open problem.

We propose a randomized algorithm for sample-efficient learning of causal graphs with confounders. We analyze the algorithm and bound the maximum number of interventional samples required to learn the causal graph with all the confounders with a given confidence level. For the causal bandit setup, we propose a two-stage approach where the first step learns a subgraph of the underlying causal graph to construct a set of POMISs, and the second phase learns the optimal arm among the POMISs. We show that the requirement of learning only a subgraph leads to significant savings in terms of interventional samples and consequently, regret. The main contributions of our work are as follows:

- We characterize the necessary and sufficient set of latent confounders in the induced subgraph on ancestors of the reward node that we need to learn/detect in order to identify all the POMISs for a causal bandit setup when the underlying causal graph is unknown.

- We propose a randomized algorithm for sample-efficient learning of causal graphs with confounders, providing theoretical guarantee on the number of interventional samples required to learn the graph with a given confidence level.

- We propose a two-phase algorithm for causal bandits with unknown causal graphs containing confounders. The first phase involves learning the induced subgraph on reward's ancestors along with a subset of latent confounders to identify all the POMISs. The next phase involves a standard bandit algorithm, e.g., upper confidence bound (UCB) algorithm. Our theoretical analysis establishes an upper bound on the cumulative regret of the overall algorithm.

## 2 Preliminaries and Problem Setup

We start with an overview of the causal bandit problem and other relevant background needed on causal models. Structural causal model (SCM) is a tuple $\mathcal{M} = \langle \mathbf{V}, \mathbf{U}, \mathbf{F}, P(\mathbf{U}) \rangle$ where $\mathbf{V} = \{V_i\}_{i=1}^n \cup \{Y\}$ is the set of observed variables, $\mathbf{U}$ is the set of independent exogenous variables, $\mathbf{F}$ is the set of deterministic structural equations and $P(\mathbf{U})$ is the distribution for exogenous variables [22]. The equations $f_i$ map the parents $(\mathsf{Pa}(V_i))$ and a subset of exogenous variables $\mathbf{U}_i \subseteq \mathbf{U}$, to the value of variable $V_i$, i.e., $V_i = f_i(\mathsf{Pa}(V_i), \mathbf{U}_i)$. We consider the causal bandit setup where all the observed variables $V_i \in \mathbf{V}$ are discrete with the domain $\Omega(V_i) = [K] := \{1, 2, 3, \ldots, K\}$, and the reward $Y$ is binary, i.e., $\Omega(Y) = \{0, 1\}$. We can associate a DAG $\mathcal{G} = (\mathbf{V}, \mathbf{E})$ with every SCM, where the vertices $\mathbf{V}$ correspond to the observed variables and edges $\mathbf{E}$ consist of directed edges $V_i \to V_j$ when $V_i \in \mathsf{Pa}(V_j)$ and bi-directed edges between $V_i$ and $V_j$ $(V_i \leftrightarrow V_j)$ when they share some common unobserved variable, also called latent confounder. We restrict ourselves to semi-Markovian causal models in which every unobserved variable has no parents and has exactly two children, both of which are observed [10]. An intervention on a set of variables $\mathbf{W} \subseteq \mathbf{V}$, denoted by $do(\mathbf{W})$, induces a post-interventional DAG ($\mathcal{G}_{\overline{\mathbf{W}}}$) with incoming edges to vertices $\mathbf{W}$ removed . We can broadly classify interventions into deterministic interventions, where variables are set to a fixed realization denoted by $do(\mathbf{W} = \mathbf{w})$, and stochastic interventions, where instead of a fixed realization we have $\mathbf{W} \sim \mathbb{P}(.)$, where $\mathbb{P}$ is a probability measure over the domain $\Omega(\mathbf{W})$. We denote the sub-model induced under hard intervention by $\mathcal{M}_{\mathbf{W}=\mathbf{w}}$ and the one induced under stochastic intervention by $\mathcal{M}_{\mathbf{W}}$. In the context of causal bandits, an arm or action corresponds to hard intervention on a subset of variables other than the reward. The goal of the agent is to identify the intervention that maximizes the expected reward. The performance of an agent is measured in terms of cumulative regret $R_T$.

$$R_T := T \max_{\mathbf{W} \subseteq \mathbf{V}} \max_{\mathbf{w} \in [K]^{|\mathbf{W}|}} \mathbb{E}[Y|do(\mathbf{W} = \mathbf{w})] - \sum_{t=1}^{T} \mathbb{E}[Y|do(\mathbf{W}_t = \mathbf{w}_t)], \tag{1}$$

where $do(\mathbf{W}_t = \mathbf{w}_t)$ represents the intervention selected by the agent in round $t$. We use the notation $\Delta_{do(\mathbf{w})}$ to define the sub-optimality gap of the corresponding arm $do(\mathbf{W} = \mathbf{w})$. We denote the descendants, ancestors and children of a vertex $V_i$ by $\mathsf{De}(V_i)$, $\mathsf{An}(V_i)$ and $\mathsf{Ch}(V_i)$ respectively. We use the notation $\mathsf{Bi}(V_i, \mathcal{G})$ to denote the set of vertices having bidirected edges to $V_i$ except the reward node $Y$. We refer to the induced graph between observed variables as the observable graph. The **transitive closure** of a graph, denoted by $\mathcal{G}^{tc}$, encodes the ancestral relationship in $\mathcal{G}$. That is, the directed edge $V_i \to V_j$ is included in $\mathcal{G}^{tc}$ only when $V_i \in \mathsf{An}(V_j)$. The **transitive reduction**, denoted by $Tr(\mathcal{G}) = (\mathbf{V}, \mathbf{E}^r)$, is a graph with the minimum number of edges such that the transitive closure is the same as $\mathcal{G}$. The connected component (c-component) of the DAG $\mathcal{G}$, containing vertex $V_i$, is denoted by $\mathsf{CC}(V_i)$, which is the maximal set of all vertices in $\mathcal{G}$ that have a path to $V_i$, consisting only of bi-directed edges [23]. For a subset of vertices $\mathbf{W} \subseteq \mathbf{V}$, we define $\mathsf{CC}(\mathbf{W}) := \bigcup_{W_i \in \mathbf{W}} \mathsf{CC}(W_i)$. In a DAG, a subset of nodes $\mathbf{W}$ d-separates two nodes $V_i$ and $V_j$ when it effectively blocks all paths between them, denoted as $V_i \perp\!\!\!\perp_d V_j | \mathbf{W}$. Blocking is a graphical criterion associated with $d$-separation [22]. A probability distribution is said to be faithful to a graph if and only if every conditional independence (CI) statement can be inferred from d-separation statements in the graph. Faithfulness is a commonly used assumption in the existing work on causal discovery [14, 24]. We assume that the following form of the interventional faithfulness assumption holds in our setup.

**Assumption 2.1.** *Consider a set of nodes $\mathbf{W} \subseteq \mathbf{V}$ and the stochastic intervention $do(\mathbf{W}, \mathbf{U})$ on $\mathbf{W}$ and any set $\mathbf{U} \subseteq \mathbf{V} \setminus \mathbf{W}$. The conditional independence (CI) statement $(\mathbf{X} \perp\!\!\!\perp \mathbf{Y} \mid \mathbf{Z})_{\mathcal{M}_{\mathbf{W}, \mathbf{U}}}$ holds in the induced model if and only if there is a corresponding d-separation statement in post-interventional graph $(\mathbf{X} \perp\!\!\!\perp_d \mathbf{Y} \mid \mathbf{Z})_{\mathcal{G}_{\overline{\mathbf{W}, \mathbf{U}}}}$, where $\mathbf{X}$, $\mathbf{Y}$, and $\mathbf{Z}$ are disjoint subsets of $\mathbf{V} \setminus \mathbf{W}$. The CI statements in the induced model are with respect to the post-interventional joint probability distribution.*

## 3 Possibly Optimal Arms in Causal Bandits with Unknown Causal Graph

The optimal intervention in a causal bandit setup is not restricted to the parent set of the reward node when the reward node $Y$ is confounded with any node in its ancestors $\mathsf{An}(Y)$ [5]. For instance, consider SCM $X_1 = U_1$ and $X_2 = X_1 \oplus U_2$ and reward $Y = X_2 \oplus U_2$, where $U_1 \sim Ber(0.5)$ and $U_2 \sim Ber(0.5)$. Note that $X_2$ and reward $Y$ are confounded in this SCM. The optimal intervention in this case is $do(X_1 = 1)$ since $\mathbb{E}[Y|do(X_1 = 1)] = 1$. The intervention on the parent of the reward $(\mathsf{Pa}(Y) = X_2)$ is suboptimal because $\mathbb{E}[Y|do(X_2 = 0)] = \mathbb{E}[Y|do(X_2 = 1)] = 0.5$. The example

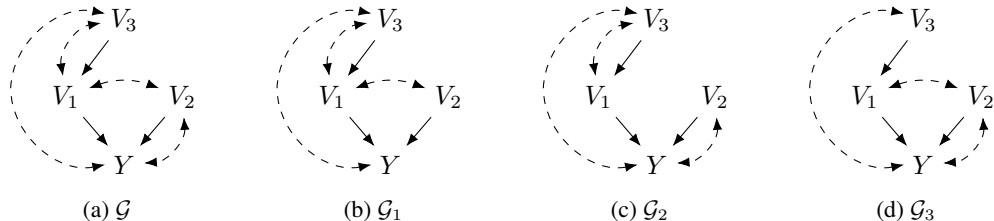

<p align="center">(a) $\mathcal{G}$     (b) $\mathcal{G}_1$     (c) $\mathcal{G}_2$     (d) $\mathcal{G}_3$</p>

Figure 1: True Causal Graph $\mathcal{G}$ with four other graphs each with one missing bi-directed edge.

shows that it is possible to construct SCMs where optimal intervention is on ancestors of the reward node instead of parents when reward node is confounded with one of its ancestors. The authors in [5] propose a graphical criterion to enumerate the set of all possibly optimal arms, which they refer to as POMISs. We revisit some definitions and results from their work.

**Definition 3.1.** *(Unobserved Confounder (UC)-Territory[5]) Consider a causal graph $\mathcal{G}(\mathbf{V}, \mathbf{E})$ with a reward node $Y$ and let $\mathcal{H}$ be $\mathcal{G}[\mathsf{An}(Y)]$. A set of variables $\mathbf{T} \subseteq V(\mathcal{H})$ containing $Y$ is called an **UC-territory** on $\mathcal{G}$ with respect to $Y$ if $De_{\mathcal{H}}(\mathbf{T}) = \mathbf{T}$ and $\mathsf{CC}_{\mathcal{H}}(\mathbf{T}) = \mathbf{T}$.*

A UC-territory is minimal if none of its subsets are UC-territories. A minimal UC-territory denoted by $\mathrm{MUCT}(\mathcal{G}, Y)$, can be constructed by extending a set of variables, starting from the reward $\{Y\}$, alternatively updating the set with the c-component and descendants of the set until there is no change.

**Definition 3.2.** *(Interventional Border)[5] Let $T$ be a minimal UC-territory on $\mathcal{G}$ with respect to $Y$. Then, $X = \mathsf{Pa}(T) \setminus T$ is called an interventional border for $\mathcal{G}$ w.r.t. $Y$ denoted by $IB(\mathcal{G}, Y)$.*

**Lemma 3.1.** *[5] For causal graph $\mathcal{G}$ with reward $Y$, $IB(\mathcal{G}_{\overline{W}}, Y)$ is a POMIS, for any $\mathbf{W} \subseteq \mathbf{V} \setminus \{Y\}$.*

Although the graphical characterization in Lemma 3.1 provides a means to enumerate the complete set of POMISs, it comes with exponential time complexity. The authors also propose an efficient algorithm for enumerating all POMISs in [5]. However, this requires knowing the true causal graph, and without it, one has to consider interventions on all possible subsets of nodes, which are exponentially many. One naive approach to tackle the problem is to learn the full causal graph with all confounders to list all POMISs. However, a question arises: *Do we need to learn/detect all possible confounders since the goal is to find POMISs and not the full graph?*

Before answering the above question, we start with an example considering the causal graphs in Figure 1. Using Lemma 3.1, the set of POMISs for the true graph $\mathcal{G}$ is $\mathcal{I}_{\mathcal{G}} = \{\phi, \{V_1\}, \{V_2\}, \{V_3\}, \{V_1, V_2\}\}$. However, for $\mathcal{G}_1$ which has the bidirected edge $V_2 \leftrightarrow Y$ missing, the set of POMISs is $\mathcal{I}_{\mathcal{G}_1} = \{\phi, \{V_2\}, \{V_1, V_2\}\}$. Also for $\mathcal{G}_2$ which has the bidirected edge $V_1 \leftrightarrow V_2$ missing, the set of POMISs is $\mathcal{I}_{\mathcal{G}_2} = \{\phi, \{V_1\}, \{V_2\}, \{V_1, V_2\}\}$. In both cases, we miss at least one POMIS, and since it is possible to construct an SCM compatible with the true causal graph $\mathcal{G}$ where any arm in POMIS is optimal, if this arm is not learned, we can suffer linear regret [5]. Although the graph $\mathcal{G}_3$ has the bidirected edge $V_1 \leftrightarrow V_3$ missing, it still has the same set of POMISs as the true graph, i.e., $\mathcal{I}_{\mathcal{G}_3} = \{\phi, \{V_1\}, \{V_2\}, \{V_3\}, \{V_1, V_2\}\}$. This example shows that only a subset of latent confounders affect the POMISs learned from the graph. We formally prove that it is necessary and sufficient to learn/detect all latent variables between the reward and its ancestors because missing any one of them will cause us to miss at least one of POMISs leading to linear regret for some bandit instances.

**Lemma 3.2.** *It is necessary to learn/detect the latent confounders between reward node $Y$ and any node $X \in \mathsf{An}(Y)$ in causal graph $\mathcal{G}$ to learn all the POMISs correctly and hence avoid linear regret.*

**Theorem 3.1.** *Consider a causal graph $\mathcal{G}(\mathbf{V}, \mathbf{E})$ and another causal graph $\mathcal{G}'$ such that they have the same vertex set and directed edges but differ in bidirected edges, with the bidirected edges in $\mathcal{G}'$ being a subset of the bidirected edges in $\mathcal{G}$. The graphs will yield different collections of POMISs if and only if there exists some $Z \in \mathsf{An}(Y)$ such that either (a) or (b) is true:*

    *(a) There is a bi-directed edge between $Z$ and $Y$ in $\mathcal{G}$ but not in $\mathcal{G}'$.*

    *(b) Neither of the graphs $\mathcal{G}'$ and $\mathcal{G}$ have a bidirected edge between $Z$ and $Y$, and there exists a bidirected edge in $\mathcal{G}$ between some $X \in MUCT(\mathcal{G}'_{\overline{\mathsf{Pa}(Z)}, \mathsf{Bi}(Z, \mathcal{G}')}, Y)$ and $Z$ but not in $\mathcal{G}'$.*

We extend Lemma 3.2 to provide necessary and sufficient conditions in Theorem 3.1 characterizing all the latent variables that need to be learned, ensuring that the POMISs learned from a sparser causal

graph match all those in the true causal graph. Suppose we have access to the induced observable subgraph $\mathcal{G}'$ on ancestors of the reward node. We can start by testing for latent confounders between $Y$ and any node in $\mathsf{An}(Y)$. Then, we need to test for latent confounders between any pair $Z \in \mathsf{An}(Y)$ such that $Z$ and $Y$ don't have a bi-directed edge between them, and $X \in \mathsf{MUCT}(\mathcal{G}'_{\overline{\mathsf{Pa}(Z),\mathsf{Bi}(Z,\mathcal{G}')}}, Y)$ until there are no new pairs to test. Theorem 3.1 can be useful because depending on the underlying causal graph, it saves us the number of latent confounders we need to test. For instance, consider a causal graph that has the reward $Y$ with $n$ different parent nodes, i.e., $\mathsf{Pa}(Y) = \{V_1, V_2, \ldots, V_n\}$, with no edges between the parents. In cases where every parent of $Y$ is confounded with $Y$, or when none of them is confounded with $Y$, we only need to test for $|\mathsf{An}(Y)|$ latent variables, as implied by Theorem 3.1. However, in the worst-case scenario, we would need to test $\binom{|\mathsf{An}(Y)|+1}{2}$ latent variables when the true graph only has the confounders $V_1 \leftrightarrow Y$ and $V_i \leftrightarrow V_{i+1}$ for all $i = 1, .., n-1$. The exact number of latents we need to test can range from $|\mathsf{An}(Y)|$ to $\binom{|\mathsf{An}(Y)|+1}{2}$ depending on the true graph. One issue still remains: we need a sample-efficient algorithm to learn the induced observable graph over $\mathsf{An}(Y)$ and to test the presence of confounders, which is addressed in upcoming sections.

## 4 Finite Sample Causal Discovery Algorithm

In this section, we propose a sample-efficient algorithm to learn causal graphs with latent confounders. We propose a two-phase approach. In the first phase, the algorithm learns the observable graph structure, i.e., the induced graph between observed variables. In the second phase, it detects the latent confounders. In the next section, we use the proposed discovery algorithm to construct the algorithm for causal bandits with an unknown graph. We begin by proposing two Lemmas to learn the ancestrality relations and latent confounders using interventions.

**Lemma 4.1.** *Consider a causal graph $\mathcal{G}(\boldsymbol{V}, \boldsymbol{E})$ and $\boldsymbol{W} \subseteq \boldsymbol{V}$. Furthermore, let $X, T \in \boldsymbol{V} \setminus \boldsymbol{W}$ be any two variables. Under the faithfulness Assumption 2.1 $(X \in \mathsf{An}(T))_{\mathcal{G}_{\overline{\boldsymbol{w}}}}$ if and only if for any $\boldsymbol{w} \in [K]^{|\boldsymbol{W}|}$, we have $P(t|do(\boldsymbol{w})) \neq P(t|do(\boldsymbol{w}), do(x))$ for some $x, t \in [K]$.*

**Lemma 4.2.** *Consider two variables $X_i$ and $X_j$ such that $X_j \notin \mathsf{An}(X_i)$ and a set of variables $(\mathsf{Pa}(X_i) \cup \mathsf{Pa}(X_j) \setminus \{X_i\}) \subseteq \boldsymbol{W}$ and $X_i, X_j \notin \boldsymbol{W}$. Under the faithfulness Assumption 2.1 there is latent confounder between $X_i$ and $X_j$ if and only if for any $\boldsymbol{w} \in [K]^{|\boldsymbol{W}|}$, we have $P(x_j \mid do(x_i), do(\boldsymbol{W} = \boldsymbol{w})) \neq P(x_j \mid x_i, do(\boldsymbol{W} = \boldsymbol{w}))$ for some realization $x_i, x_j \in [K]$.*

These Lemmas are modified versions of Lemma 1 in [14] and Interventional Do-see test in [14], respectively. The difference between Lemma 4.1 and Lemma 1 in [14] is that we have an inequality test that can be used in the sample-efficient discovery instead of a statistical independence test. The Interventional Do-see test in [14] is valid for adjacent nodes only; however, our Lemma 4.2 can be used to test presence of latent confounder between any pair of nodes. This is because the condition in Lemma 4.2, $X_j \notin \mathsf{An}(X_i)$, can always be satisfied for any pair by flipping the order when one node is an ancestor of the other. In order to provide theoretical guarantees on sampling complexity, the inequality conditions are not enough; we need to assume certain gaps similar to [7, 9, 17].

**Assumption 4.1.** *Consider a causal graph $\mathcal{G}(\boldsymbol{V}, \boldsymbol{E})$ and $\boldsymbol{W} \subseteq \boldsymbol{V}$. Furthermore, let $X, T \in \boldsymbol{V} \setminus \boldsymbol{W}$ be any two variables. Then, we have $(X \in \mathsf{An}(T))_{\mathcal{G}_{\overline{\boldsymbol{w}}}}$ if and only if for any $\boldsymbol{w} \in [K]^{|\boldsymbol{W}|}$, we have $|P(t|do(\boldsymbol{w})) - P(t|do(\boldsymbol{w}), do(x))| > \epsilon$ for some $x, t \in [K]$, where $\epsilon > 0$ is some constant.*

**Assumption 4.2.** *Consider two variables $X_i$ and $X_j$ such that $X_j \notin \mathsf{An}(X_i)$ and a set of variables $(\mathsf{Pa}(X_i) \cup \mathsf{Pa}(X_j) \setminus \{X_i\}) \subseteq \boldsymbol{W}$ and $X_i, X_j \notin \boldsymbol{W}$. There is a latent confounder or a bidirected edge between $X_i$ and $X_j$ if and only if for any $\boldsymbol{w} \in [K]^{|\boldsymbol{W}|}$, we have $\big|P(x_j \mid do(x_i), do(\boldsymbol{W} = \boldsymbol{w})) - P(x_j \mid x_i, do(\boldsymbol{W} = \boldsymbol{w}))\big| > \gamma$ for some realization $x_i, x_j \in [K]$ and some constant $\gamma > 0$.*

### 4.1 Learning the Observable Graph

We propose Algorithm 1 to learn the transitive closure under any arbitrary intervention $do(\boldsymbol{W})$, denoted by $\mathcal{G}^{tc}_{\overline{\boldsymbol{W}}}$. We use the Assumption 4.1 to bound the number of samples for ancestrality tests. We start with an empty graph and add edges by running ancestrality tests for all pairs of nodes in $\boldsymbol{V} \setminus \boldsymbol{W}$, resulting in the transitive closure $\mathcal{G}^{tc}_{\overline{\boldsymbol{W}}}$. We recall that the transitive reduction $Tr(\mathcal{G}) = (\boldsymbol{V}, \boldsymbol{E}^r)$ of a DAG $\mathcal{G} = (\boldsymbol{V}, \boldsymbol{E})$ is unique, with $\boldsymbol{E}^r \subseteq \boldsymbol{E}$, and it can be computed in polynomial time [25]. Also, note that $Tr(\mathcal{G}) = Tr(\mathcal{G}^{tc})$. We propose a randomized Algorithm 2 similar to the one proposed in

---

**Algorithm 1:** Learn the Transitive Closure of the Causal Graph under any intervention, i.e., $\mathcal{G}_{\overline{\mathbf{W}}}^{tc}$

---
**1** **Function** LearnTransitiveClosure($\mathbf{V}, \mathbf{W}, \delta_1, \delta_2$):
**2**    $\mathbf{E} = \emptyset$ , Fix some $\mathbf{w} \in [K]^{|\mathbf{W}|}$ and A = $max(\frac{8}{\epsilon^2}, \frac{8}{\gamma^2}) \log \frac{2nK^2}{\delta_1}$ and B = $\frac{8}{\epsilon^2} \log \frac{2nK^2}{\delta_2}$
**3**    Get $B$ samples from $do(\mathbf{W} = \mathbf{w})$
**4**    Get $A$ samples from every $do(X_i = x_i, \mathbf{W} = \mathbf{w}) \, \forall X_i \in \mathbf{V} \setminus \mathbf{W}$ and $\forall x_i \in [K]$
**5**    **for** *every pair $X_i, X_j \in \mathbf{V} \setminus \mathbf{W}$* **do**
**6**       Use the Interventional Data to Test if $(X_i \in \mathsf{An}(X_j))_{\mathcal{G}_{\overline{\mathbf{W}}}}$
**7**       **if** $\exists\, x_i, x_j \in [K] \; s.t. \; |\widehat{P}(x_j|do(\mathbf{w})) - \widehat{P}(x_j|do(\mathbf{w}), do(x_i))| > \frac{\epsilon}{2}$ **then**
**8**          $\mathbf{E} \longleftarrow \mathbf{E} \cup (X_i, X_j)$

**9**    **return** The graph's transitive closure $(\mathbf{V}, \mathbf{E})$ and All Interventional data
**10** **End Function**

---

**Algorithm 2:** Learn the Observable Graph

---
**1** **Function** LearnObservableGraph($\mathbf{V}, \alpha, d_{max}, \delta_1, \delta_2$):
**2**    $\mathbf{E} = \emptyset$ & $\mathcal{I}Data = \emptyset$
**3**    **for** $i = 1 \, : \, 8\alpha \, d_{max} \, \log(n)$ **do**
**4**       $\mathbf{W} = \emptyset$
**5**       **for** $V_i \in \mathbf{V}$ **do**
**6**          $\mathbf{W} \leftarrow \mathbf{W} \cup V_i$ with probability $1 - \frac{1}{2d_{max}}$
**7**       $\mathcal{G}_{\overline{\mathbf{W}}}^{tc}, \text{Data}_{\overline{\mathbf{W}}} = $ LearnTransitiveClosure($\mathbf{V}, \mathbf{W}, \delta_1, \delta_2$)
**8**       Compute the transitive reduction $Tr(\mathcal{G}_{\overline{\mathbf{W}}}^{tc})$ & add any missing edges from $Tr(\mathcal{G}_{\overline{\mathbf{W}}}^{tc})$ to $\mathbf{E}$
**9**       $\mathcal{I}Data = \mathcal{I}Data \cup \text{Data}_{\overline{\mathbf{W}}}$ (Keep Saving Interventional Data)
**10**    **return** The observable graph structure $(\mathbf{V}, \mathbf{E})$ and interventional data samples in $\mathcal{I}Data$
**11** **End Function**

---

[14] that repeatedly uses Algorithm 1 to learn the observable graph structure. The motivation behind the randomized Algorithm 2 is Lemma 5 from [14], which states that for any edge $(X_i, X_j)$, consider a set of variables $\mathbf{W}$ such that $\{W_i : \pi(W_i) > \pi(X_i) \,\&\, W_i \in \mathsf{Pa}(X_j)\} \subseteq \mathbf{W}$ where $\pi$ is any total order that is consistent with the partial order implied by the DAG, i.e., $\pi(X) < \pi(Y)$ iff $X \in \mathsf{An}(Y)$. In this case, the edge $(X_i, X_j)$ will be present in the graph $Tr(\mathcal{G}_{\overline{\mathbf{W}}})$. Algorithm 2 randomly selects $\mathbf{W}$, computes the transitive reduction of the post-interventional graphs, and finally accumulates all edges found in the transitive reduction across iterations. Algorithm 2 takes a parameter $d_{\max}$, which must be greater than or equal to the highest graph degree for our theoretical guarantees to hold.

**Lemma 4.3.** *Suppose that the Assumption 4.1 holds and we have access to $max(\frac{8}{\epsilon^2}, \frac{8}{\gamma^2}) \log \frac{2K^2}{\delta_1}$ samples from $do(X_i = x_i, \mathbf{W} = \mathbf{w}) \, \forall x_i \in [K]$ and $\frac{8}{\epsilon^2} \log \frac{2K^2}{\delta_2}$ samples from $do(\mathbf{W} = \mathbf{w})$ for a fixed $w \in [K]^{|\mathbf{W}|}$ and $\mathbf{W} \subseteq \mathbf{V}$. Then, with probability at least $1 - \delta_1 - \delta_2$, we have $(X_i \in \mathsf{An}(X_j))_{\mathcal{G}_{\overline{\mathbf{W}}}}$ if and only if $\exists\, x_i, x_j \in [K] \; s.t. \; \left|\widehat{P}(x_j \mid do(\mathbf{w})) - \widehat{P}(x_j \mid do(\mathbf{w}), do(x_i))\right| > \frac{\epsilon}{2}$.*

Lemma 4.3 provides the sample complexity for running ancestrality tests. Algorithm 1 selects a realization $w \in [K]^{|\mathbf{W}|}$, takes $B$ samples from the intervention $do(\mathbf{W} = \mathbf{w})$, and $A$ samples from every $do(X_i = x_i, \mathbf{W} = \mathbf{w})$ for all $X_i \in \mathbf{V} \setminus \mathbf{W}$ and $x_i \in [K]$ interventions. Thus, in the worst case, Algorithm 1 requires $KAn + B$ samples to learn the true transitive closure with high probability. We formally prove this result in the Lemma 4.4.

**Lemma 4.4.** *Algorithm 1 learns the true transitive closure under any intervention, i.e., $\mathcal{G}_{\overline{\mathbf{W}}}^{tc}$, with probability at least $1 - n\delta_1 - \delta_2$ with a maximum $KAn + B$ interventional samples. If we set $\delta_1 = \frac{\delta}{2n}$ and $\delta_2 = \frac{\delta}{2}$, then Algorithm 1 learns true transitive closure with probability at least $1 - \delta$.*

Algorithm 2 repeatedly invokes Algorithm 1 to learn $Tr(\mathcal{G}_{\overline{\mathbf{W}}})$ for randomly sampled $\mathbf{W}$. Through this iterative process, it accumulates edges across iterations, ultimately constructing the observable graph structure. To establish the sampling complexity guarantee for Algorithm 2, we leverage the result from Lemma 4.4. The Theorem 4.1 gives the sampling complexity for learning the true observable graph with high probability.

**Theorem 4.1.** *Algorithm 2 learns the true observable graph with probability at least* $1 - \frac{1}{n^{\frac{\alpha}{2d_{max}}-2}} - 8\alpha d_{max} \log(n)(n\delta_1 + \delta_2)$ *with* $8\alpha d_{max} \log n (KAn + B)$ *interventional samples. If we set* $\alpha = \frac{2d_{max}\log(\frac{2}{\delta}+2)}{\log n}$, $\delta_1 = \frac{\delta}{32\alpha d_{max} n \log n}$ *and* $\delta_2 = \frac{\delta}{32\alpha d_{max}\log n}$, *then Algorithm 2 learns the true observable graph with probability at least of* $1 - \delta$. *(We have* $A = \max\left(\frac{8}{\epsilon^2}, \frac{8}{\gamma^2}\right)\log\frac{2nK^2}{\delta_1}$ & $B = \frac{8}{\epsilon^2}\log\frac{2nK^2}{\delta_2}$ *as in line 2 of Algorithm 1.)*

## 4.2 Learning the Latent Confounders

Assumption 4.2 can be used to test for latents between any pair of observed variables. Note that while using Algorithm 2, we save and return all the interventional data samples. These samples can be reused to detect latent confounders in the next phase. For any variables $X_i$ and $X_j$ such that $X_j \notin \mathsf{An}(X_i)$, we need access to interventional samples $do(\mathbf{W} = \mathbf{w})$ such that $(\mathsf{Pa}(X_i) \cup \mathsf{Pa}(X_j) \setminus \{X_i\}) \subseteq \mathbf{W}$ and $X_i$ & $X_j \notin \mathbf{W}$. In the supplementary material, we demonstrate that randomly selecting the target set $\mathbf{W}$ in Algorithm 2 ensures that we have access to all such datasets for all pairs of observed variables with high probability. In addition to simple causal effects we need to estimate the conditional causal effect of the form $P(x_j|x_i, do(\mathbf{W} = \mathbf{w}))$. To bound the number of samples required to ensure accurate estimation of the conditional causal effects, we rely on Assumption 4.3. Note that Assumption 4.3 does not restrict the applicability of our algorithm; it simply assumes that under an intervention $do(\mathbf{W} = \mathbf{w})$, either the probability of observing a realization $X_i = x_i$ is zero or is lower-bounded by some constant $\eta > 0$. The role of this assumption is to bound the number of interventional samples required for accurate estimation of the conditional causal effects.

**Assumption 4.3.** *For any variable* $X_i \in \mathbf{V}$ *and any intervention* $do(\mathbf{W} = \mathbf{w})$ *where* $\mathbf{W} \subseteq \mathbf{V}$ *and* $\mathbf{w} \in [K]^{|\mathbf{W}|}$, *we assume that either* $P(x_i|do(\mathbf{W} = \mathbf{w})) = 0$ *or* $P(x_i|do(\mathbf{W} = \mathbf{w})) \geq \eta > 0$.

---

**Algorithm 3:** Learn the Causal Graph along-with the Latent Confounders

1 **Function** LearnCausalGraph($\alpha, d_{max}, \delta_1, \delta_2, \delta_3, \delta_4$):
2     $\mathcal{G}, \mathcal{I}Data = $ LearnObservableGraph($\alpha, d_{max}, \delta_1, \delta_2$)
3     $C = \frac{16}{\eta\gamma^2}\log(\frac{2n^2K^2}{\delta_3}) + \frac{1}{2\eta^2}\log(\frac{2n^2K^2}{\delta_4})$, $B = \frac{8}{\epsilon^2}\log\frac{2nK^2}{\delta_2}$
4     **for** *every pair* $X_i, X_j \in \mathbf{V}$ **do**
5         If $X_j \in \mathsf{An}(X_i)$, swap them.
6         Find interventional data sets $do(\mathbf{W} = \mathbf{w})$ and $do(X_i = x_i, \mathbf{W} = \mathbf{w})$ from $\mathcal{I}Data$ s.t. $(\mathsf{Pa}(X_i) \cup \mathsf{Pa}(X_j) \setminus \{X_i\}) \subseteq \mathbf{W}$ and $X_i$ & $X_j \notin \mathbf{W}$
7         Get $\max(0, C - B)$ new samples for $do(\mathbf{W} = \mathbf{w})$
8         **if** $\exists\, x_i, x_j \in [K]$ s.t. $|\widehat{P}(x_j|do(x_i), do(\mathbf{w})) - \widehat{P}(x_j|x_i, do(\mathbf{w}))| > \frac{\gamma}{2}$ **then**
9             Add bi-directed edge $X_i \leftrightarrow X_j$ to graph $\mathcal{G}$
10     **return** The Causal Graph with Latent Confounders $\mathcal{G}$
11 **End Function**

---

**Lemma 4.5.** *Consider two nodes* $X_i$ *and* $X_j$ *s.t.* $X_j \notin \mathsf{An}(X_i)$ *and suppose that Assumptions 2.1 4.2 hold and we have access to* $\max(\frac{8}{\epsilon^2}, \frac{8}{\gamma^2})\log\frac{2K^2}{\delta_1}$ *samples from* $do(X_i = x_i, \mathbf{W} = \mathbf{w})$ $\forall x_i \in [K]$ *and* $\frac{16}{\eta\gamma^2}\log(\frac{2K^2}{\delta_3}) + \frac{1}{2\eta^2}\log(\frac{2K^2}{\delta_4})$ *samples from* $do(\mathbf{W} = \mathbf{w})$ *for a fixed* $w \in [K]^{|\mathbf{W}|}$ *and* $\mathbf{W} \subseteq \mathbf{V}$ *such that* $(\mathsf{Pa}(X_i) \cup \mathsf{Pa}(X_j) \setminus \{X_i\}) \subseteq \mathbf{W}$ *and* $X_i$ & $X_j \notin \mathbf{W}$. *Then, with probability at least* $1 - \delta_1 - \delta_3 - \delta_4$, *we have a latent confounder between* $X_i$ *and* $X_j$ *iff* $\exists\, x_i, x_j \in [K]$ *s.t.* $\left| \widehat{P}(x_j|do(x_i), do(\mathbf{w})) - \widehat{P}(x_j|x_i, do(\mathbf{w})) \right| > \frac{\gamma}{2}$.

Lemma 4.5 establishes the sample complexity for detecting the presence of latent confounders for any pair of nodes in the causal graph. Using results from Theorem 4.1 and Lemma 4.5, we bound the number of interventions required by the proposed Algorithm 3 to learn the causal graph along with the latent confounders. Theorem 4.2 provides the sample complexity guarantee for Algorithm 3 to learn the true causal graph, including all latent confounders, with a given confidence level. An important feature of the sampling complexity result in Theorem 4.2 is that the number of intervention samples needed to learn the causal graph scales polynomially with the number of nodes $n$.

**Theorem 4.2.** *Algorithm 3 learns the true causal graph with latents with probability at least* $1 - \frac{2}{n^{\frac{\alpha}{2d_{max}}-2}} - 8\alpha d_{max} \log(n)(n\delta_1 + (\delta_2 + \delta_3 + \delta_4))$ *with a maximum of* $8\alpha d_{max} \log n (KAn + \max(B,C))$ *interventional samples. If we set* $\alpha = \frac{2d_{max}\log\left(\frac{4}{\delta}+2\right)}{\log n}$, $\delta_1 = \frac{\delta}{64\alpha d_{max} n \log n}$ *and* $\delta_2 = \delta_3 = \delta_4 = \frac{\delta}{64\alpha d_{max}\log n}$, *then Algorithm 3 learns the true causal graph with probability at least* $1 - \delta$. *(A and B are given by line 2 of Algorithm 1 and C is given by line 3 of Algorithm 3.)*

Suppose the constant gaps $\epsilon$ and $\gamma$ in Assumptions 4.1 and 4.2 are close; then, we have $C > \frac{1}{\eta}A \geq \frac{1}{\eta}B$. The value of the constant $0 < \eta < 1$ is usually small in practical scenarios, so the quantity $C$ is much greater than both $B$ or $A$. This implies that the number of samples required to test the presence of latent variables is greater than that required to learn ancestral relations. This is because we need to accurately estimate conditional causal effects to detect latent variables, which requires a large number of samples compared to simple causal effects. Theorem 3.1 is useful here because it shows that we do not need to test for confounders between all pairs of nodes among ancestors of the reward node to learn the POMIS set.

## 5 Algorithm for Causal Bandits with Unknown Graph Structure

Algorithm 4 is the sketch of our algorithm for causal bandits with unknown graph structure. The detailed algorithm with all steps explained is given in the supplementary material (Algorithm 6). Algorithm 4 first learns the transitive closure of the graph $\mathcal{G}^{tc}$ to find ancestors of the reward node $Y$. This is because POMISs are only subsets of $\mathsf{An}(Y)$. The next step is to learn the observed graph structure among the reward $Y$ and nodes in $\mathsf{An}(Y)$. Instead of detecting the presence of confounders between all pairs of nodes in $\mathsf{An}(Y)$ as in Algorithm 3, we focus on identifying the necessary and sufficient ones, as characterized by Theorem 3.1. This approach is more sample-efficient since it tests for fewer latent confounders. The exact saving in terms of samples depends on the underlying causal graph and is hard to characterize in general. The last step of Algorithm 4 is to run a simple bandit algorithm, e.g., UCB algorithm [26], to identify the optimal arm from the POMISs. Given that Assumptions 4.1, 4.2, and 4.3 hold, and the reward is binary ($Y \in \{0,1\}$), using the results from Lemma 4.4 and Theorem 4.2, we provide a worst-case regret bound for Algorithm 4 in Theorem 5.1.

---

**Algorithm 4:** Sketch of Algorithm for causal bandits with unknown graph structure

---
1   Calculate $\alpha, \delta_1, \delta_2, \delta_3, \delta_4$ as in Theorem 5.1
2   $\mathcal{G}^{tc}$ = LearnTransitiveClosure($\mathbf{W} = \phi, \frac{\delta}{2n}, \frac{\delta}{n}$)
3   $\mathcal{G}, \mathcal{ID}ata$ = LearnObservableGraph($\mathsf{An}(Y)_{\mathcal{G}^{tc}}, \alpha, d_{max}, \delta_1, \delta_2$)
4   # Learn the bi-directed edges between reward $Y$ and all nodes $X_i \in \mathsf{An}(Y)$ and update $\mathcal{G}$.
5   **for** *every* $X_i \in \mathsf{An}(Y)_{\mathcal{G}^{tc}}$ **do**
6     $\lfloor$   $\mathcal{G}$ = DetectLatentConfounder($\mathcal{G}, X_i, X_j, \delta_2, \delta_3, \delta_4, \mathcal{ID}ata$) (Algorithm 5)
7   **while** *There is a new pair that is tested* **do**
8     Find a new pair $(Z, X)$ s.t. $Z \in \mathsf{An}(Y)$ such that $Z$ and $Y$ don't have a bi-directed edge between them in $\mathcal{G}$ and $X \in \mathrm{MUCT}(\mathcal{G}_{\overline{\mathsf{Pa}(Z), \mathsf{Bi}(Z, \mathcal{G})}}, Y)$ and test for the latent and update $\mathcal{G}$.
9     $\mathcal{G}$ = DetectLatentConfounder($\mathcal{G}, Z, X, \delta_2, \delta_3, \delta_4, \mathcal{ID}ata$)
10   Learn the set of POMISs $\mathcal{I}_{\mathcal{G}}$ from the graph $\mathcal{G}$ (Using Algorithm 1 from [5]).
11   Run UCB algorithm over the arm set $A = \{\Omega(I) \mid \forall I \in \mathcal{I}_{\mathcal{G}}\}$.

---

**Theorem 5.1.** *Algorithm 4 learns the true set of POMISs with probability at least* $1 - 2\delta$. *Under the event that it learns POMISs correctly, the cumulative regret is bounded as follows:*

$$R_T \leq Kn \max\left(\frac{8}{\epsilon^2}, \frac{8}{\gamma^2}\right) \log \frac{4n^2 K^2}{\delta} \;+\; \frac{8}{\epsilon^2} \log \frac{4nK^2}{\delta} \;+$$

$$8\alpha d_{max}\left(KA|\mathsf{An}(Y)| \;+\; \max(B,C)\right) \log\left(|\mathsf{An}(Y)|\right) \;+\; \sum_{\mathbf{s} \in \{\Omega(I) | \forall I \in \mathcal{I}_{\mathcal{G}}\}} \Delta_{do(\mathbf{s})}\left(1 + \frac{\log T}{\Delta_{do(\mathbf{s})}^2}\right),$$

*where A and B are given by line 2 of Algorithm 1, and C is given by line 3 of Algorithm 3 by setting* $\alpha = \frac{2d_{max}\log\left(\frac{4}{\delta}+2\right)}{\log|\mathsf{An}(Y)|}$, $\delta_1 = \frac{\delta}{64\alpha d_{max}|\mathsf{An}(Y)|\log|\mathsf{An}(Y)|}$ *and* $\delta_2 = \delta_3 = \delta_4 = \frac{\delta}{64\alpha d_{max}\log|\mathsf{An}(Y)|}$.

The first three terms in the regret bound correspond to the interventional samples required to learn the ancestors of the reward node, and then the set of POMISs ($\mathcal{I}_{\mathcal{G}}$). The last term corresponds to the regret incurred by running the UCB algorithm over the POMIS set. The number of interventional samples used to learn the true set of POMISs, with high probability, has polynomial scaling with respect to the number of nodes $n$ in the graph. However, the total number of arms in the POMIS set, in the worst case, can exhibit exponential scaling with respect to the number of ancestors of the reward node $|\mathsf{An}(Y)|$. The advantage of sample-efficient discovery is that it helps us reduce the action space before applying the UCB algorithm. If the graph is not densely confounded, the total number of arms in the POMIS set would be small, and running causal discovery before the bandit algorithm is advantageous. Without discovery, one would always have to run the UCB or a standard MAB solver with exponentially many arms. For instance, if the causal graph has $n$ nodes, there will be $\sum_{i=1}^{n} \binom{n}{i} K^i = (K+1)^n$ different possible arms/interventions.

## 6 Experiments

Theorem 5.1 establishes the worst-case upper bound for cumulative regret when we need to test latent confounders between all pairs of nodes within $\mathsf{An}(Y)$. However, Algorithm 4 selectively examines only a subset of latent confounders sufficient to infer the true POMIS set, as outlined in Theorem 3.1. Although the advantage is hard to quantify in general, we demonstrate it using simulations on randomly generated graphs. We sample a random ordering $\sigma$ among the vertices. Then, for each $n$th node, we determine its in-degree as $X_n = \max(1, \mathsf{Bin}(n-1, \rho))$, followed by selecting its parents through uniform sampling from the preceding nodes in the ordering. Finally, we chordalize the graph using the elimination algorithm [27], employing an elimination ordering that is the reverse of $\sigma$. Additionally, we introduce a confounder between every pair of nodes with a probability of $\rho_L$. For all the simulations, we randomly sample 50 causal graphs with different values of densities $\rho$ and $\rho_L$ and assume that all variables are binary for simplicity, i.e., $K = 2$. We set the value of $\delta$ to 0.99, and the gaps $\gamma = \epsilon = 0.01$ and $\eta = 0.05$. We plot interventional samples used to learn the induced observable graph on $\mathsf{An}(Y)$ with and without latent confounders, as well as the samples required to learn the POMIS set by Algorithm 4. The width of confidence interval is set to 2 standard deviations.

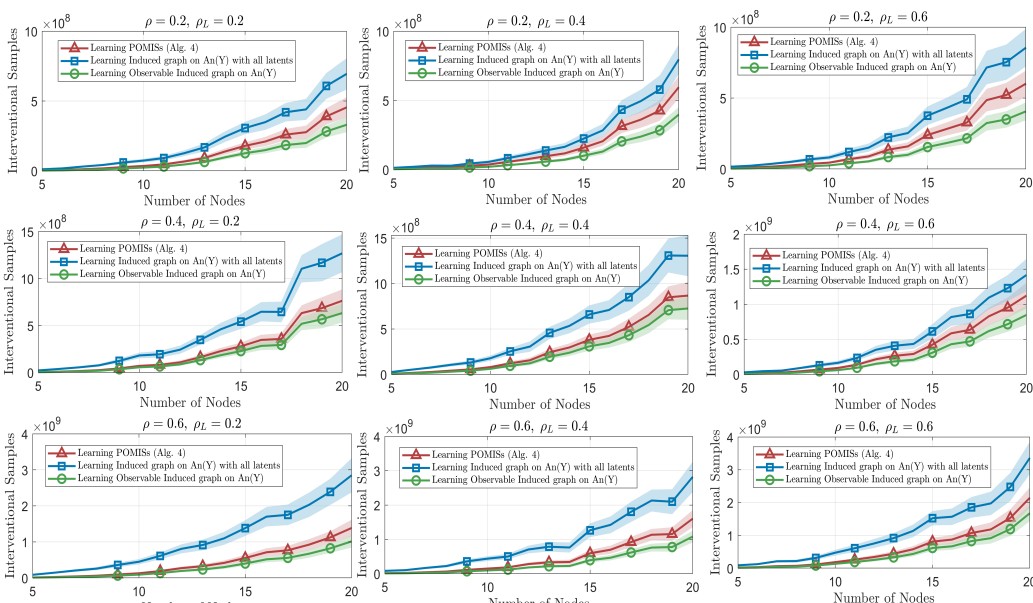

Figure 2: Simulations to demonstrate the advantage of Algorithm 4 over full graph discovery (Learning all possible latents)

The simulation results in Figure 2 demonstrate that Algorithm 4 requires fewer samples than learning the induced graph on $\mathsf{An}(Y)$, which includes all confounders. However, as $\rho_L$ increases for a fixed $\rho$, this advantage diminishes, as illustrated in Figure 2. The trend remains consistent as the density parameters $\rho$ and $\rho_L$ are varied from $0.2$, $0.4$, and $0.6$. The plots in Figure 3 compare the exponentially

growing arms in causal bandits with intervention samples used by our algorithm to learn the reduced action set in the form of POMISs. This demonstrates the major advantage of our algorithm, which, instead of exploring an exponentially large action set as in naive UCB algorithms, uses interventions to reduce the action space to the POMIS set before applying the UCB algorithm. Additionally, the number of intervention samples required in the first phase of identifying the true POMIS set grows polynomially with respect to the number of nodes in the graph. However, the number of arms in the POMIS set can still exhibit exponential scaling with respect to the number of ancestors of the reward node in the worst case.

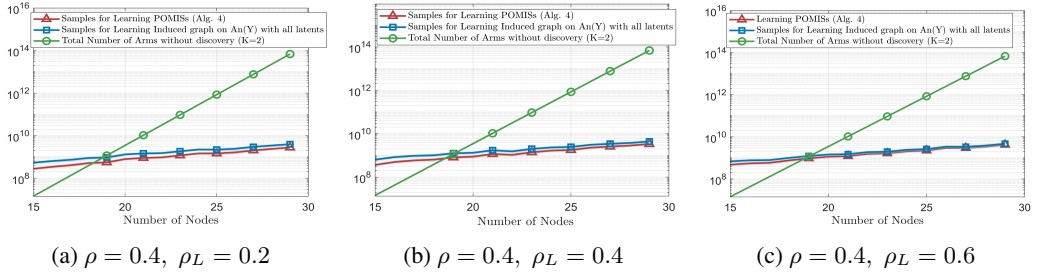

(a) $\rho = 0.4, \ \rho_L = 0.2$  (b) $\rho = 0.4, \ \rho_L = 0.4$  (c) $\rho = 0.4, \ \rho_L = 0.6$

Figure 3: Simulations to demonstrate advantage of discovery for causal bandits.

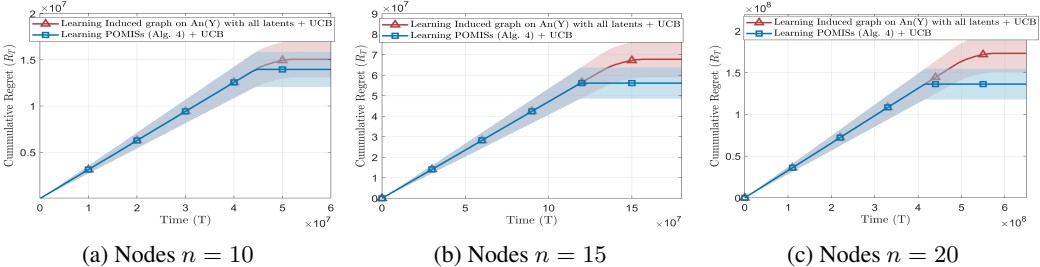

(a) Nodes $n = 10$  (b) Nodes $n = 15$  (c) Nodes $n = 20$

Figure 4: Cumulative regret for Algorithm 4 versus learning all possible latents ($\rho = \rho_L = 0.3$).

We also run the UCB algorithm on the learned POMIS set and plot the cumulative regret in Figure 4. Since the number of time steps $T$ is on the order of $10^8$, it is not feasible to store and plot cumulative regret for every time step over multiple randomly sampled graphs; therefore, we downsample the cumulative regret to show the overall trend. The downsampling, along with the large scale of the $y$-axis, makes the regret in the discovery phase appear linear with a fixed slope, although it is piece-wise linear if we zoom in. Also, the UCB phase converges very fast compared to the discovery phase because the number of POMISs for randomly sampled graphs is small. We plot the results for graphs with 10, 15, and 20 nodes, and in all cases, we can see the advantage of partial discovery compared to full discovery, since Algorithm 4 finds the POMIS set with fewer samples. The code to reproduce our experimental results is available at `https://github.com/CausalML-Lab/CausalBandits_with_UnknownGraph`.

# 7 Conclusion

We show that partial discovery is sufficient to achieve sublinear regret for causal bandits with an unknown causal graph containing latent confounders. Without relying on causal discovery, one must consider interventions on all possible subsets of nodes, which is infeasible. Therefore, we propose a two-phase approach: the first phase learns the induced subgraph of the ancestors of the reward node, along with a subset of confounders, to construct a set of possibly optimal arms. We demonstrate that the number of interventional samples in the first phase required to identify the POMIS set scales polynomially with respect to the number of nodes in the causal graph. In the next phase, we apply the Upper Confidence Bound (UCB) algorithm to the reduced action space to find the optimal arm.

# 8 Acknowledgment

Murat Kocaoglu acknowledges the support of NSF CAREER 2239375, IIS 2348717, Amazon Research Award and Adobe Research.

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

# A  Supplemental Material

## A.1  Review of $d$-separation:

Consider three disjoint sets of nodes $\mathbf{X}$, $\mathbf{Y}$, and $\mathbf{Z}$ in the causal graph $\mathcal{G} = (\mathbf{V}, \mathbf{E})$. The sets of nodes $\mathbf{X}$ and $\mathbf{Y}$ are $d$-separated given $\mathbf{Z}$, denoted by $(\mathbf{X} \perp\!\!\!\perp_d \mathbf{Y}|\mathbf{Z})_{\mathcal{G}}$, if and only if there exists no path, directed or undirected, between any node in set $\mathbf{X}$ and any node in set $\mathbf{Y}$ such that for every collider on the path, either the collider itself or one of its descendants is included in the set $\mathbf{Z}$, and no other non-collider nodes on the path are included in the set $\mathbf{Z}$. (A collider on a path is a node with both arrows converging, e.g., $B$ is a collider on the path $ABC$ in $A \to B \leftarrow C$).

## A.2  Pearl's Rules of do-Calculus ([22]):

Let $\mathcal{G}$ represent the causal DAG, and let $P$ denote the probability distribution induced by the corresponding causal model. For any disjoint subsets of variables $\mathbf{X}, \mathbf{Y}, \mathbf{Z}$, and $\mathbf{W}$, the following rules apply:

**Rule 1:** (Insertion/deletion of observations):

$$P(\mathbf{y}|do(\mathbf{x}), \mathbf{z}, \mathbf{w}) = P(\mathbf{y}|do(\mathbf{x}), \mathbf{w}) \quad \text{if} \quad (\mathbf{Y} \perp\!\!\!\perp_d \mathbf{Z}|\mathbf{X}, \mathbf{W})_{\mathcal{G}_{\overline{\mathbf{X}}}}. \tag{2}$$

**Rule 2:** (Action/observation exchange):

$$P(\mathbf{y}|do(\mathbf{x}), do(\mathbf{z}), \mathbf{w}) = P(\mathbf{y}|do(\mathbf{x}), \mathbf{z}, \mathbf{w}) \quad \text{if} \quad (\mathbf{Y} \perp\!\!\!\perp_d \mathbf{Z}|\mathbf{X}, \mathbf{W})_{\mathcal{G}_{\overline{\mathbf{X}}\underline{\mathbf{Z}}}}. \tag{3}$$

**Rule 3:** (Insertion/deletion of actions):

$$P(\mathbf{y}|do(\mathbf{x}), do(\mathbf{z}), \mathbf{w}) = P(\mathbf{y}|do(\mathbf{x}), \mathbf{w}) \quad \text{if} \quad (\mathbf{Y} \perp\!\!\!\perp_d \mathbf{Z}|\mathbf{X}, \mathbf{W})_{\mathcal{G}_{\overline{\mathbf{X}, \overline{\mathbf{Z}(\mathbf{W})}}}}, \tag{4}$$

where $\mathbf{Z}(\mathbf{W})$ is the set of nodes in $\mathbf{Z}$ that are not ancestors of any of the nodes in $\mathbf{W}$ in the graph $\mathcal{G}_{\overline{\mathbf{X}}}$.

## A.3  Function to Detect Presence of Latent Confounder:

---
**Algorithm 5:** Function to Detect Presence of Latent Confounder

---
1 **Function** DetectLatentConfounder($\mathcal{G}, X_i, X_j, \delta_2, \delta_3, \delta_4, \mathcal{I}Data$):
2      $C = \frac{16}{\eta\gamma^2} \log(\frac{2n^2K^2}{\delta_3}) + \frac{1}{2\eta^2} \log(\frac{2n^2K^2}{\delta_4})$, $B = \frac{8}{\epsilon^2} \log \frac{2nK^2}{\delta_2}$
3      if $X_j \in \mathsf{An}(X_i)$ swap them.
4      Find interventional data sets $do(\mathbf{W} = \mathbf{w})$ and $do(X_i = x_i, \mathbf{W} = \mathbf{w})$ from $\mathcal{I}Data$ s.t.
       $(\mathsf{Pa}(X_i) \cup \mathsf{Pa}(X_j) \setminus \{X_i\}) \subseteq \mathbf{W}$ and $X_i$ & $X_j \notin \mathbf{W}$
5      Get $\max(0, B - C)$ new samples for $do(\mathbf{W} = \mathbf{w})$
6      **if** $\exists x_i, x_j \in [K]$ s.t. $|\widehat{P}(x_j|do(x_i), do(\mathbf{w})) - \widehat{P}(x_j|x_i, do(\mathbf{w}))| > \frac{\gamma}{2}$ **then**
7          Add bi-directed edge $X_i \leftrightarrow X_j$ to graph $\mathcal{G}$
8      **return** Updated Causal Graph $\mathcal{G}$
9 **End Function**

---

## A.4  Proof of Lemma 3.2:

**Lemma. 3.2:** *It is necessary to learn/detect the latent confounders between reward node $Y$ and any node $X \in \mathsf{An}(Y)$ in causal graph $\mathcal{G}$ to learn all the POMISs correctly and hence avoid linear regret.*

Before proceeding to the proof, we recall an important result from [5]: For a causal graph $\mathcal{G}$ with reward variable $Y$, $\mathrm{IB}(\mathcal{G}_{\overline{\mathbf{W}}}, Y)$ is a POMIS for any $\mathbf{W} \subseteq \mathbf{V} \setminus Y$.

**Proof:** Consider a causal graph $\mathcal{G}(\mathbf{V}, \mathbf{E})$ with a node $X \in \mathsf{An}(Y)$ such that there exists a latent confounder between $X$ and the reward $Y$. Suppose we do not detect the presence of the confounder and have access to another causal graph $\mathcal{G}'$ with everything the same as $\mathcal{G}$ except that there is no confounder between $X$ and $Y$. We show that there exists one such POMIS that we cannot learn from $\mathcal{G}'$, which actually exists in the true causal graph $\mathcal{G}$. To prove this, consider a set of nodes

$\mathbf{W} = \mathsf{Pa}(X) \cup \mathsf{Ch}(\mathsf{Pa}(X)) \cup \mathsf{CC}(X) \setminus \{X, Y\}$. For the graph $\mathcal{G}'$, note that $X \notin \mathrm{MUCT}(\mathcal{G}'_{\overline{\mathbf{W}}}, Y)$, and also there $\nexists Z \in \mathsf{Ch}(\mathsf{Pa}(X)) \setminus \{X\}$ s.t. $Z \in \mathrm{MUCT}(\mathcal{G}'_{\overline{\mathbf{W}}}, Y)$. This implies that $\nexists Z \in \mathsf{Pa}(X)$ s.t. $Z \in \mathrm{IB}(\mathcal{G}'_{\overline{\mathbf{W}}}, Y)$. However, for the true graph $\mathcal{G}$, we have a different $\mathrm{IB}(\mathcal{G}_{\overline{\mathbf{W}}}, Y)$ for the same definition of $\mathbf{W}$ because it contains the bi-directed edge between $X$ and $Y$, which implies that $X \in \mathrm{MUCT}(\mathcal{G}_{\overline{\mathbf{W}}}, Y)$, and as a result, $\mathsf{Pa}(X) \subseteq \mathrm{IB}(\mathcal{G}_{\overline{\mathbf{W}}}, Y)$. Also, in the case $\mathsf{Pa}(X) = \emptyset$, we have a different POMIS. On this side, note that $X \notin \mathrm{MUCT}(\mathcal{G}'_{\overline{\mathbf{W}}}, Y)$, which implies that along the causal path from $X$ to $Y$, there must be one node $Z$ such that $Z \in \mathrm{MUCT}(\mathcal{G}'_{\overline{\mathbf{W}}}, Y)$, which implies either $X$ or one of its descendants on the path from $X$ to $Y$ is in $\mathrm{IB}(\mathcal{G}'_{\overline{\mathbf{W}}}, Y)$, which is not the case for $\mathcal{G}$ since $X \in \mathrm{MUCT}(\mathcal{G}_{\overline{\mathbf{W}}}, Y)$. Thus, we have different interventional boundary or POMIS for the two causal graphs $\mathcal{G}$ and $\mathcal{G}'$ given the above choice of $\mathbf{W}$, even if $X$ has no parents.

The next step is to show that the particular POMIS $\mathrm{IB}(\mathcal{G}_{\overline{\mathbf{W}}}, Y)$ cannot be learned from the DAG $\mathcal{G}'$, i.e., $\mathrm{IB}(\mathcal{G}_{\overline{\mathbf{W}}}, Y) \neq \mathrm{IB}(\mathcal{G}'_{\overline{\mathbf{W}'}}, Y)$ for any $\mathbf{W}' \subseteq \mathbf{V}$. We need to show this because of the graphical characterization of POMISs in Lemma 3.1. Using the definition of $\mathbf{W}$, note that $\mathsf{Pa}(X) \subseteq \mathrm{IB}(\mathcal{G}_{\overline{\mathbf{W}}}, Y)$ and for all $Z \in \mathsf{Ch}(\mathsf{Pa}(X)) \setminus \{X\}$, there exists either $Z \in \mathrm{IB}(\mathcal{G}_{\overline{\mathbf{W}}}, Y)$ or $\mathsf{De}(Z) \setminus \{Y\} \in \mathrm{IB}(\mathcal{G}_{\overline{\mathbf{W}}}, Y)$. Also, if there are such nodes in $\mathsf{CC}(X) \setminus \{X, Y\}$ which do not have a path to $X$ comprised of directed edges only, call such set of nodes $T$. If $T \neq \phi$, then for all $t \in T$, we have either $t \in \mathrm{IB}(\mathcal{G}_{\overline{\mathbf{W}}}, Y)$ or $\mathsf{De}(t) \setminus \{Y\} \in \mathrm{IB}(\mathcal{G}_{\overline{\mathbf{W}}}, Y)$. Also, note that $\nexists Z \in \mathsf{De}(X) \cup \{X\}$ such that $Z \in \mathrm{IB}(\mathcal{G}_{\overline{\mathbf{W}}}, Y)$. Now consider DAG $\mathcal{G}'$ with the bi-directed edge between $X$ and $Y$ missing. Assume by contradiction $\exists \mathbf{W}' \subseteq \mathbf{V}$ such that $\mathrm{IB}(\mathcal{G}_{\overline{\mathbf{W}}}, Y) = \mathrm{IB}(\mathcal{G}'_{\overline{\mathbf{W}'}}, Y)$. This, however, using the aforementioned characterization of $\mathrm{IB}(\mathcal{G}_{\overline{\mathbf{W}}}, Y)$ implies that $\nexists Z \in \mathsf{Ch}(\mathsf{Pa}(X)) \setminus \{X\}$ such that $Z \in \mathrm{MUCT}(\mathcal{G}'_{\overline{\mathbf{W}'}}, Y)$ and also $\nexists t \in T$ such that $t \in \mathrm{MUCT}(\mathcal{G}'_{\overline{\mathbf{W}'}}, Y)$ using the aforementioned definition of $T$. However, note that we need $\mathsf{Pa}(X) \subseteq \mathrm{IB}(\mathcal{G}'_{\overline{\mathbf{W}'}}, Y)$, which under the given choice of $\mathbf{W}$ is only possible when $X \in \mathrm{MUCT}(\mathcal{G}'_{\overline{\mathbf{W}'}}, Y)$, which would require is a bi-directed edge between $X$ and $Y$ in the DAG $\mathcal{G}'$, which is a contradiction. Also, for the case when $\mathsf{Pa}(X) = \phi$, we have a contradiction because we require the following to be true: $\nexists Z \in \mathsf{De}(X) \cup \{X\}$ such that $Z \in \mathrm{IB}(\mathcal{G}'_{\overline{\mathbf{W}'}}, Y)$. For the given choice of $\mathbf{W}$, it implies that there is a bi-directed edge between $X$ and $Y$ in the DAG $\mathcal{G}'$, which is again a contradiction. Thus, by contradiction, we show that $\nexists \mathbf{W}' \subseteq \mathbf{V}$ such that $\mathcal{G}'$, i.e., $\mathrm{IB}(\mathcal{G}_{\overline{\mathbf{W}}}, Y) \neq \mathrm{IB}(\mathcal{G}'_{\overline{\mathbf{W}'}}, Y)$. This implies that we will miss at least one POMIS if we do not learn or detect latent confounders between the reward node $Y$ and any node $X \in \mathsf{An}(Y)$, and may incur linear regret. This completes the proof of Lemma 3.2.

## A.5 Proof of Theorem 3.1:

Before proving Theorem 3.1, we state and prove another Lemma. We then extend this Lemma to prove Theorem 3.1.

**Lemma A.1.** *Consider a causal graph $\mathcal{G}(\mathbf{V}, \mathbf{E})$ and another graph $\mathcal{G}'$ such that they have the same vertex set and directed edges but differ in bidirected edges, with the bidirected edges in $\mathcal{G}'$ being a subset of the bidirected edges in $\mathcal{G}$. The graphs will yield different collections of POMISs if there exists some $Z \in \mathsf{An}(Y)$ such that either (a) or (b) is true:*

  *(a) There is a bi-directed edge between $Z$ and $Y$ in $\mathcal{G}$ but not in $\mathcal{G}'$.*

  *(b) Neither of the graphs $\mathcal{G}'$ and $\mathcal{G}$ have a bidirected edge between $Z$ and $Y$, and there exists a bidirected edge in $\mathcal{G}$ between some $X \in MUCT(\mathcal{G}'_{\overline{\mathsf{Pa}(Z), \mathsf{Bi}(Z, \mathcal{G}')}}, Y)$ and $Z$ but not in $\mathcal{G}'$.*

**Proof:** The first half of Lemma A.1, i.e., "The graphs will yield different collections of POMISs if there exists some $Z \in \mathsf{An}(Y)$ such that there is a bi-directed edge between $Z$ and $Y$ in $\mathcal{G}$ but not in $\mathcal{G}'$," is the same as Lemma 3.2, and the same proof applies here. The reason is that in graph $\mathcal{G}'$, we miss a latent variable between reward and one of its ancestors, which was actually present in the true graph $\mathcal{G}$. We only need to proof the second half of Lemma A.1 i.e. graphs will yield different collections of POMISs if there exists some $Z \in \mathsf{An}(Y)$ such that (b) is true. Consider a causal graph $\mathcal{G}(\mathbf{V}, \mathbf{E})$ and another DAG $\mathcal{G}'$ such that they have the same vertex set and directed edges, but differ in bi-directed edges. Consider a causal graph $\mathcal{G}(\mathbf{V}, \mathbf{E})$ and another DAG $\mathcal{G}'$ such that they have the same vertex set and directed edges, but differ in bi-directed edges. We show that if neither of the graphs $\mathcal{G}'$ and $\mathcal{G}$ have a bidirected edge between $Z$ and $Y$, and there exists a bidirected edge in $\mathcal{G}$ between some $X \in \mathrm{MUCT}(\mathcal{G}'_{\overline{\mathsf{Pa}(Z), \mathsf{Bi}(Z, \mathcal{G}')}}, Y)$ and $Z$, then there exists one such POMIS that we cannot learn from $\mathcal{G}'$, which actually exists in the true causal graph $\mathcal{G}$. To prove this, consider a set of nodes $\mathbf{W} = \mathsf{Pa}(Z) \cup \mathsf{Ch}(\mathsf{Pa}(Z) \setminus \mathsf{An}(X)) \cup \mathsf{Bi}(Z, \mathcal{G}') \setminus \{X, Z, Y\}$. For

the graph $\mathcal{G}'$, note that $Z \notin \text{MUCT}(\mathcal{G}'_{\overline{\mathbf{W}}}, Y)$, and also there $\nexists N \in \text{Ch}(\text{Pa}(Z) \setminus \text{An}(X)) \setminus \{Z\}$ s.t. $N \in \text{MUCT}(\mathcal{G}'_{\overline{\mathbf{W}}}, Y)$. This implies that $\nexists N \in \text{Pa}(Z) \setminus \text{An}(X)$ s.t. $N \in \text{IB}(\mathcal{G}'_{\overline{\mathbf{W}}}, Y)$. However, for the true graph $\mathcal{G}$, we have a different $\text{IB}(\mathcal{G}_{\overline{\mathbf{W}}}, Y)$ for the same definition of $\mathbf{W}$ because it contains the bi-directed edge between $X$ and $Z$, which implies that $Z \in \text{MUCT}(\mathcal{G}_{\overline{\mathbf{W}}}, Y)$, and as a result, $\text{Pa}(Z) \setminus \text{An}(X) \subseteq \text{IB}(\mathcal{G}_{\overline{\mathbf{W}}}, Y)$. Also, in the case $\text{Pa}(Z) \setminus \text{An}(X) = \emptyset$, we have different a POMIS. On this side, note that $Z \notin \text{MUCT}(\mathcal{G}'_{\overline{\mathbf{W}}}, Y)$, which implies that along the causal path from $Z$ to $Y$, there must be one node $N$ such that $N \in \text{MUCT}(\mathcal{G}'_{\overline{\mathbf{W}}}, Y)$, which implies either $Z$ or one of its descendants on the path from $Z$ to $Y$ is in $\text{IB}(\mathcal{G}'_{\overline{\mathbf{W}}}, Y)$, which is not the case for $\mathcal{G}$ since $Z \in \text{MUCT}(\mathcal{G}_{\overline{\mathbf{W}}}, Y)$. Thus, we have different interventional boundary or POMIS for the two causal graphs $\mathcal{G}$ and $\mathcal{G}'$ given the above choice of $\mathbf{W}$.

The next step is to show that the particular POMIS $\text{IB}(\mathcal{G}_{\overline{\mathbf{W}}}, Y)$ cannot be learned from the DAG $\mathcal{G}'$, i.e., $\text{IB}(\mathcal{G}_{\overline{\mathbf{W}}}, Y) \neq \text{IB}(\mathcal{G}'_{\overline{\mathbf{W}'}}, Y)$ for any $\mathbf{W}' \subseteq \mathbf{V}$. We need to show this because of the graphical characterization of POMISs in Lemma 3.1. Using the definition of $\mathbf{W}$, note that $\text{Pa}(Z) \setminus \text{An}(X) \subseteq \text{IB}(\mathcal{G}_{\overline{\mathbf{W}}}, Y)$ and for all $N \in \text{Ch}(\text{Pa}(Z) \setminus \text{An}(X)) \setminus \{Z\}$, there exists either $N \in \text{IB}(\mathcal{G}_{\overline{\mathbf{W}}}, Y)$ or $\text{De}(N) \setminus \{Y\} \in \text{IB}(\mathcal{G}_{\overline{\mathbf{W}}}, Y)$. Also, if there are such nodes in $\text{Bi}(Z, \mathcal{G}') \setminus \{X, Z, Y\}$ which do not have a path to $Z$ comprising of directed edges only, call such set of nodes $T$. If $T \neq \phi$, then for all $t \in T$, we have either $t \in \text{IB}(\mathcal{G}_{\overline{\mathbf{W}}}, Y)$ or $\text{De}(t) \setminus \{Y\} \in \text{IB}(\mathcal{G}_{\overline{\mathbf{W}}}, Y)$. Also, note that $\nexists N \in \text{De}(Z) \cup \{Z\}$ such that $N \in \text{IB}(\mathcal{G}_{\overline{\mathbf{W}'}}, Y)$. Now consider the DAG $\mathcal{G}'$ with the bi-directed edge between $Z$ and $Y$ missing. Assume by contradiction $\exists \mathbf{W}' \subseteq \mathbf{V}$ such that $\text{IB}(\mathcal{G}_{\overline{\mathbf{W}}}, Y) = \text{IB}(\mathcal{G}'_{\overline{\mathbf{W}'}}, Y)$. This, however, using the aforementioned characterization of $\text{IB}(\mathcal{G}_{\overline{\mathbf{W}}}, Y)$ implies that $\nexists N \in \text{Ch}(\text{Pa}(Z) \setminus \text{An}(X)) \setminus \{Z\}$ such that $N \in \text{MUCT}(\mathcal{G}'_{\overline{\mathbf{W}'}}, Y)$ and also $\nexists t \in T$ such that $t \in \text{MUCT}(\mathcal{G}'_{\overline{\mathbf{W}'}}, Y)$ for the aforementioned definition of $T$. However, note that we need $\text{Pa}(Z) \setminus \text{An}(X) \subseteq \text{IB}(\mathcal{G}'_{\overline{\mathbf{W}'}}, Y)$, which under the given choice of $\mathbf{W}$ is only possible when $Z \in \text{MUCT}(\mathcal{G}'_{\overline{\mathbf{W}'}}, Y)$, which would require a bi-directed edge between $Z$ and $X$ in the DAG $\mathcal{G}'$, which is a contradiction. Also, for the case when $\text{Pa}(Z) = \phi$, we have a contradiction because we require the following to be true: $\nexists N \in \text{De}(Z) \cup \{Z\}$ such that $N \in \text{IB}(\mathcal{G}'_{\overline{\mathbf{W}'}}, Y)$. For the given choice of $\mathbf{W}$, it implies that there is a bi-directed edge between $Z$ and $X$ in the DAG $\mathcal{G}'$, which is again a contradiction. Thus, by contradiction, we show that $\nexists \mathbf{W}' \subseteq \mathbf{V}$ such that $\mathcal{G}'$, i.e., $\text{IB}(\mathcal{G}_{\overline{\mathbf{W}}}, Y) \neq \text{IB}(\mathcal{G}'_{\overline{\mathbf{W}'}}, Y)$. This implies that we miss atleast one POMIS when either of statements (a) and (b) hold. This completes the proof of Lemma A.1.

We now proceed to the formal proof for Theorem 3.1:

**Theorem. 3.1:** *Consider a causal graph $\mathcal{G}(\mathbf{V}, \mathbf{E})$ and another DAG $\mathcal{G}'$ such that they have the same vertex set and directed edges but differ in bidirected edges, with the bidirected edges in $\mathcal{G}'$ being a subset of the bidirected edges in $\mathcal{G}$. The graphs will yield different collections of POMISs if and only if there exists some $Z \in \text{An}(Y)$ such that either (a) or (b) is true:*

    *(a) There is a bi-directed edge between $Z$ and $Y$ in $\mathcal{G}$ but not in $\mathcal{G}'$ .*

    *(b) Neither of the graphs $\mathcal{G}'$ and $\mathcal{G}$ have a bidirected edge between $Z$ and $Y$, and there exists a bidirected edge in $\mathcal{G}$ between some $X \in \text{MUCT}(\mathcal{G}'_{\overline{\text{Pa}(Z), \text{Bi}(Z, \mathcal{G}')}}, Y)$ and $Z$ but not in $\mathcal{G}'$.*

**Proof:** One direction for Theorem 3.1 is proved already in Lemma A.1. We only to need to prove the other direction which is that two causal graphs $\mathcal{G}$ and $\mathcal{G}'$ such that they have the same vertex set and directed edges, but differ in bi-directed edges will yield same collections POMISs when neither of statements (a) and (b) is true. Note when neither of (a) or (b) is true the graphs $\mathcal{G}$ and $\mathcal{G}'$ might still have a different set of bi-directed edges. We will have two possible scenarios here. Suppose $\mathcal{G}$ has a bi-directed edge between some $Z \in \text{An}(Y)$ and some $X \in \text{An}(Y)$, such that there is a bi-directed edge between pair of vertices $(Z, Y)$ and $(X, Y)$ in both the graphs and the bi-directed edge between $X$ and $Z$ is absent in $\mathcal{G}'$. Further, assume neither of statements (a) and (b) hold. In this case, despite the absence of a bi-directed edge between $X$ and $Z$ in $\mathcal{G}'$, the graphs will yield the same set of POMISs. This is because $Z \notin MUCT(\mathcal{G}_{\overline{\mathbf{W}}}, Y)$ for some set of nodes $\mathbf{W}$ only when $Z \in \mathbf{W}$, and the same is the case for $\mathcal{G}$ because they share a bi-directed edge between $Z$ and $Y$. By symmetry, we have the argument hold for $X$ as well. So, the presence or absence of bi-directed edges between $X$ and $Z$ does not change the set of POMISs learned from the graph when both $X$ and $Z$ are confounded with reward $Y$ already. Thus, we can delete all such bi-directed edges one by one from $\mathcal{G}$ while the set of POMISs learned from each of the intermediate causal graphs stays the same. Consider the second scenario, where $\mathcal{G}$ has bi-directed edges between a node $Z \in \text{An}(Y)$, such that there is no bi-directed

edge between $Z$ and $Y$ in both graphs ($\mathcal{G}$ and $\mathcal{G}'$) and a node $X$ that has the following characteristics: $X \in \text{MUCT}(\mathcal{G}'_{\overline{\mathbf{W}}}, Y)$ for some set $\mathbf{W} \subseteq \mathbf{V}$ but $X \notin \text{MUCT}(\mathcal{G}'_{\overline{\text{Pa}(Z), \text{Bi}(Z, \mathcal{G}')}}, Y)$. However, the bi-directed edge between $X$ and $Z$ is absent in $\mathcal{G}'$. Further, assume neither of statements (a) and (b) hold. The condition $X \in \text{MUCT}(\mathcal{G}'_{\overline{\mathbf{W}}}, Y)$ but $X \notin \text{MUCT}(\mathcal{G}'_{\overline{\text{Pa}(Z), \text{Bi}(Z, \mathcal{G}')}}, Y)$ implies that either $\exists N \in \text{Pa}(Z)$ such that $N \in \text{MUCT}(\mathcal{G}'_{\overline{\mathbf{W}}}, Y)$ or $\exists N \in \text{Bi}(Z, \mathcal{G}')$ such that $N \in \text{MUCT}(\mathcal{G}'_{\overline{\mathbf{W}}}, Y)$. Since bi-directed edges in $\mathcal{G}'$ are a subset of bi-directed edges in $\mathcal{G}$, we have: Either $\exists N \in \text{Pa}(Z)$ such that $N \in \text{MUCT}(\mathcal{G}_{\overline{\mathbf{W}}}, Y)$ or $\exists N \in \text{Bi}(Z, \mathcal{G})$ such that $N \in \text{MUCT}(\mathcal{G}_{\overline{\mathbf{W}}}, Y)$. Note that any MUCT is closed under the $\text{De}(.)$ and $\text{CC}(.)$ operations, i.e., for any MUCT, say $\mathbf{T}$, we have $\text{De}(\mathbf{T}) = \mathbf{T}$ and $\text{CC}(\mathbf{T}) = \mathbf{T}$. if $\exists N \in \text{Pa}(Z)$ such that $N \in \text{MUCT}(\mathcal{G}_{\overline{\mathbf{W}}}, Y)$ or $\exists N \in \text{Bi}(Z, \mathcal{G})$ such that $N \in \text{MUCT}(\mathcal{G}_{\overline{\mathbf{W}}}, Y)$, we already have $Z \in \text{MUCT}(\mathcal{G}_{\overline{\mathbf{W}}}, Y)$ using the definition of MUCT. The bi-directed edge between $X$ and $Z$ will play a role only when $\nexists N \in \text{Pa}(Z)$ such that $N \in \text{MUCT}(\mathcal{G}_{\overline{\mathbf{W}}}, Y)$ and $\nexists N \in \text{Bi}(Z, \mathcal{G})$ such that $N \in \text{MUCT}(\mathcal{G}_{\overline{\mathbf{W}}}, Y)$ for any choice of $\mathbf{W}$. Recall that the given condition $X \in \text{MUCT}(\mathcal{G}'_{\overline{\mathbf{W}}}, Y)$ but $X \notin \text{MUCT}(\mathcal{G}'_{\overline{\text{Pa}(Z), \text{Bi}(Z, \mathcal{G}')}}, Y)$ already implies that either $\exists N \in \text{Pa}(Z)$ such that $N \in \text{MUCT}(\mathcal{G}_{\overline{\mathbf{W}}}, Y)$ or $\exists N \in \text{Bi}(Z, \mathcal{G})$ such that $N \in \text{MUCT}(\mathcal{G}_{\overline{\mathbf{W}}}, Y)$. Thus absence or presence of bi-directed edge between $X$ and $Z$ will have no effect on POMISs learned from graph $\mathcal{G}$ in this scenario as well. Combining both of the scenarios when neither of the conditions of (a) and (b) hold, all other bi-directed edges from $\mathcal{G}$, which are absent in $\mathcal{G}'$, can be removed one by one from $\mathcal{G}$ while keeping the POMISs learned from both the intermediate graphs the same. Since $\mathcal{G}$ and $\mathcal{G}'$ only differ in bi-directed edges, with bi-directed edges in $\mathcal{G}'$ being a subset of those in $\mathcal{G}$, eventually both graphs will become identical, which proves the statement: Two graphs $\mathcal{G}$ and $\mathcal{G}'$ will have the same POMISs if neither of the statements (a) or (b) hold true. This completes the proof of the Theorem 3.1.

### A.6   Proof of Lemma 4.1:

Consider a causal graph $\mathcal{G}(\mathbf{V}, \mathbf{E})$ and $\mathbf{W} \subseteq \mathbf{V}$. Furthermore, let $X, T \in \mathbf{V} \setminus \mathbf{W}$ be any two variables. Fix some realization $\mathbf{w} \in [K]^{|\mathbf{W}|}$. Under post interventional faithfulness Assumption 2.1 we want to prove: $(X \in \text{An}(T))_{\mathcal{G}_{\overline{\mathbf{w}}}} \iff P(t|do(\mathbf{w})) \neq P(t|do(\mathbf{w}), do(x))$ for some $x, t \in [K]$.

**Forward Direction ( $\implies$ ):** $(X \in \text{An}(T))_{\mathcal{G}_{\overline{\mathbf{w}}}} \implies P(t|do(\mathbf{w})) \neq P(t|do(\mathbf{w}), do(x))$ for some $x, t \in [K]$. By contradiction, assume $P(t|do(\mathbf{w})) = P(t|do(\mathbf{w}), do(x)), \forall x, t \in [K]$. This implies that $P(t|do(\mathbf{w}), do(x)) = P(t|do(\mathbf{w})) = $ some function of only $t$ and $\mathbf{w}$. This implies that for the sub-model $\mathcal{M}_{\mathbf{W}, X}$ the following CI statements holds: $(T \perp\!\!\!\perp X)_{\mathcal{M}_{\mathbf{w}, X}}$. However, note that if $(X \in \text{An}(T))_{\mathcal{G}_{\overline{\mathbf{w}}}}$, then we still have $(X \in \text{An}(T))_{\mathcal{G}_{\overline{\mathbf{w}, X}}}$. This implies there is a directed path from $X$ to $T$ in the post-interventional graph $\mathcal{G}_{\overline{\mathbf{W}, X}}$. Therefore, we have: $(T \not\perp\!\!\!\perp_d X)_{\mathcal{G}_{\overline{\mathbf{w}, X}}}$. Note that under the post interventional faithfulness Assumption 2.1, the CI statement $(T \perp\!\!\!\perp X)_{\mathcal{M}_{\mathbf{w}, X}}$ can hold only if the $d$-separation statement holds $(T \perp\!\!\!\perp_d X)_{\mathcal{G}_{\overline{\mathbf{w}, X}}}$, which is clearly a contradiction. This completes the proof for the forward direction.

**Reverse Direction ( $\impliedby$ ):** $(X \in \text{An}(T))_{\mathcal{G}_{\overline{\mathbf{w}}}} \impliedby P(t|do(\mathbf{w})) \neq P(t|do(\mathbf{w}), do(x))$ for some $x, t \in [K]$. We prove the contrapositive statement instead, i.e., $(X \notin \text{An}(T))_{\mathcal{G}_{\overline{\mathbf{w}}}} \implies P(t|do(\mathbf{w})) = P(t|do(\mathbf{w}), do(x)), \forall x, t \in [K]$. Note that $(X \notin \text{An}(T))_{\mathcal{G}_{\overline{\mathbf{w}}}}$ clearly implies that $(X \notin \text{An}(T))_{\mathcal{G}_{\overline{\mathbf{w}, X}}}$ which implies that $(T \perp\!\!\!\perp_d X)_{\mathcal{G}_{\overline{\mathbf{w}, X}}}$. Thus, using Rule 3 of Pearl's do calculus, we have: $P(t|do(\mathbf{w}), do(x)) = P(t|do(\mathbf{w})), \forall x, t \in [K]$. This completes the proof of the reverse direction.

### A.7   Proof of Lemma 4.2:

Consider two variables $X_i$ and $X_j$ such that $X_j \notin \text{An}(X_i)$ and a set of variables $(\text{Pa}(X_i) \cup \text{Pa}(X_j) \setminus \{X_i\}) \subseteq \mathbf{W}$ and $X_i \& X_j \notin \mathbf{W}$. Fix some realization $\mathbf{w} \in [K]^{|\mathbf{W}|}$. Under the post-interventional faithfulness Assumption 2.1 we want to show that: There is latent confounder between $X_i$ and $X_j$ $\iff P(x_j \mid do(x_i), do(\mathbf{W} = \mathbf{w})) \neq P(x_j \mid x_i, do(\mathbf{W} = \mathbf{w}))$ for some realization $x_i, x_j \in [K]$.

**Forward Direction ( $\implies$ ):** There is latent confounder between $X_i$ and $X_j$ such that $X_j \notin \text{An}(X_i)$ $\implies P(x_j \mid do(x_i), do(\mathbf{W} = \mathbf{w})) \neq P(x_j \mid x_i, do(\mathbf{W} = \mathbf{w}))$ for some realization $x_i, x_j \in [K]$. By contradiction assume $P(x_j \mid do(x_i), do(\mathbf{W} = \mathbf{w})) = P(x_j \mid x_i, do(\mathbf{W} = \mathbf{w})) \forall x_i, x_j \in [K]$. Recall

that: $X_j = f_j(\mathsf{Pa}(X_j), \mathbf{U_j})$. Since there is latent confounder between $X_i$ and $X_j$ call it $L_{ij}$. Also note that $L_{ij} \in \mathbf{U_j}$. Define $\mathbf{U'_j} := \mathbf{U_j} \setminus \{L_{i,j}\}$

$$P(x_j \mid do(x_i), do(\mathbf{W})) = P(x_j \mid do(x_i), do(\mathsf{pa}(X_i)), do(\mathsf{pa}(X_j) \setminus \{x_i\}))) \tag{5}$$

where the interventions $do(\mathsf{Pa}(X_i))$ and $do(\mathsf{Pa}(X_j)))$ are consistent with $do(x_i)$ and $do(\mathbf{W} = \mathbf{w})$. The equation 5 holds by the application of Pearl's do-calculus Rule 3 because, by definition of the set $\mathbf{W}$, we have $(\mathsf{Pa}(X_i) \cup \mathsf{Pa}(X_j) \setminus \{X_i\}) \subseteq \mathbf{W}$ and $X_i, X_j \notin \mathbf{W}$. All the extra intervention targets can simply be deleted, and we are left with intervention on $X_i$, $\mathsf{Pa}(X_i)$, and $\mathsf{Pa}(X_j)$.

$$P(x_j \mid do(x_i), do(\mathbf{W})) = \sum_{\mathbf{u'_j}, \, l_{i,j}} P(x_j \mid do(x_i), do(\mathsf{pa}(X_i)), do(\mathsf{pa}(X_j) \setminus \{x_i\}), \mathbf{U'}_j = \mathbf{u'}_j, L_{ij} = l_{ij})$$
$$\times P(\mathbf{U'}_j = \mathbf{u'}_j, L_{ij} = l_{ij}) \tag{6}$$

We have another application of Pearl's do-calculus Rule 3 because interventions on observed variables don't affect unobserved variables, as there are no causal/directed paths from observed to unobserved variables. Also we have:

$$P(x_j \mid x_i, do(\mathbf{W})) = P(x_j \mid x_i, do(\mathsf{pa}(X_i)), do(\mathsf{pa}(X_j) \setminus \{x_i\}))) \tag{7}$$

The equation 7 holds by the application of Pearl's do-calculus Rule 3 because, by definition of the set $\mathbf{W}$, we have $(\mathsf{Pa}(X_i) \cup \mathsf{Pa}(X_j) \setminus \{X_i\}) \subseteq \mathbf{W}$ and $X_i, X_j \notin \mathbf{W}$. All the extra intervention targets can simply be deleted, and we are left with conditioning on $X_i = x_i$ and interventions on $\mathsf{Pa}(X_i)$ and $\mathsf{Pa}(X_j)$.

$$P(x_j \mid x_i, do(\mathbf{W})) = \sum_{\mathbf{u'_j}, \, l_{i,j}} P(x_j \mid x_i, do(\mathsf{pa}(X_i)), do(\mathsf{pa}(X_j) \setminus \{x_i\}), \mathbf{U'}_j = \mathbf{u'}_j, L_{ij} = l_{ij})$$
$$\times P(\mathbf{U'}_j = \mathbf{u'}_j, L_{ij} = l_{ij}|x_i, do(\mathsf{pa}(X_i)), do(\mathsf{pa}(X_j) \setminus \{x_i\})) \tag{8}$$

Using Pearl's do-calculus Rule 2, we can replace the conditioning $X_i = x_i$ with the intervention $do(x_i)$ in $P(x_j \mid x_i, do(\mathsf{pa}(X_i)), do(\mathsf{pa}(X_j) \setminus \{x_i\}), \mathbf{U'}_j = \mathbf{u'}_j, L_{ij} = l_{ij})$ because $X_j \notin \mathsf{An}(X_i)$ and $\mathsf{Pa}(X_i)$ are already intervened on. Also, the latent confounder $L_{ij}$ is conditioned on, so there is no open backdoor path from $X_i$ to $X_j$. Thus, we have:

$$P(x_j \mid x_i, do(\mathbf{W})) = \sum_{\mathbf{u'_j}, \, l_{i,j}} P(x_j \mid do(x_i), do(\mathsf{pa}(X_i)), do(\mathsf{pa}(X_j) \setminus \{x_i\}), \mathbf{U'}_j = \mathbf{u'}_j, L_{ij} = l_{ij})$$
$$\times P(\mathbf{U'}_j = \mathbf{u'}_j, L_{ij} = l_{ij}|x_i, do(\mathsf{pa}(X_i)), do(\mathsf{pa}(X_j) \setminus \{x_i\})) \tag{9}$$

From the Equations 6 and 9 and assumption $P(x_j \mid do(x_i), do(\mathbf{W} = \mathbf{w})) = P(x_j \mid x_i, do(\mathbf{W} = \mathbf{w}))$ $\forall x_i, x_j \in [K]$ we have:

$$\sum_{\mathbf{u'_j}, \, l_{i,j}} P(x_j \mid do(x_i), do(\mathsf{pa}(X_i)), do(\mathsf{pa}(X_j) \setminus \{x_i\}), \mathbf{U'}_j = \mathbf{u'}_j, L_{ij} = l_{ij})$$
$$\times \left( P(\mathbf{U'}_j = \mathbf{u'}_j, L_{ij} = l_{ij}|x_i, do(\mathsf{pa}(X_i)), do(\mathsf{pa}(X_j) \setminus \{x_i\})) - P(\mathbf{U'}_j = \mathbf{u'}_j, L_{ij} = l_{ij}) \right) = 0 \tag{10}$$

Since probabilities are non-negative, whenever $P(x_j \mid do(x_i), do(\mathsf{pa}(X_i)), do(\mathsf{pa}(X_j) \setminus \{x_i\}), \mathbf{U'}_j = \mathbf{u'}_j, L_{ij} = l_{ij}) > 0$, we must have:

$$P(\mathbf{U'}_j = \mathbf{u'}_j, L_{ij} = l_{ij}|x_i, do(\mathsf{pa}(X_i)), do(\mathsf{pa}(X_j) \setminus \{x_i\})) = P(\mathbf{U'}_j = \mathbf{u'}_j, L_{ij} = l_{ij}). \quad (11)$$

However, since we know that $L_{ij}$ is a confounder between $X_i$ and $X_j$, we have an edge $L_{ij} \rightarrow X_i$ in the causal graph, which implies that under any intervention $do(\mathbf{Z})$ such that $X_i \notin \mathbf{Z}$, we must have $(L_{ij} \not\perp\!\!\!\perp X_i)_{\mathcal{M}_\mathbf{Z}}$ by interventional faithfulness Assumption 2.1. This implies that there exists a realization $x_i^*$ and $l_{ij}^*$ such that:

$$P(\mathbf{U'}_j = \mathbf{u'}_j, L_{ij} = l_{ij}^*|x_i^*, do(\mathsf{pa}(X_i)), do(\mathsf{pa}(X_j) \setminus \{x_i\})) \neq P(\mathbf{U'}_j = \mathbf{u'}_j, L_{ij} = l_{ij}^*) \quad (12)$$

Now, using the combination $do(\mathbf{W} = \mathbf{w})$ and a special choice of realizations $x_i^*$ and $l_{ij}^*$, we must have at least one special realization $x_j^*$ such that: $P(x_j^* \mid do(x_i^*), do(\mathsf{Pa}(X_i)), do(\mathsf{Pa}(X_j) \setminus \{x_i\}), \mathbf{U'}_j = \mathbf{u'}_j, L_{ij} = l_{ij}^*) > 0$. Combining this with Equations 12 and 10, we conclude for some $x_i^*, x_j^* \in [K]$, we have $P(x_j^* \mid do(x_i^*), do(\mathbf{W} = \mathbf{w})) \neq P(x_j^* \mid x_i^*, do(\mathbf{W} = \mathbf{w}))$. Thus this leads to contradiction. Thus if there is a latent confounder between $X_i$ and $X_j \implies P(x_j \mid do(x_i), do(\mathbf{W} = \mathbf{w})) \neq P(x_j \mid x_i, do(\mathbf{W} = \mathbf{w}))$ for some realization $x_i, x_j \in [K]$. This completes the proof of the forward direction.

**Reverse Direction** ( $\impliedby$ ): For a pair of variables $X_i$ and $X_j$ such that $X_j \notin \mathsf{An}(X_i)$, if $P(x_j \mid do(x_i), do(\mathbf{W} = \mathbf{w})) \neq P(x_j \mid x_i, do(\mathbf{W} = \mathbf{w}))$ for some realizations $x_i, x_j \in [K]$, then there is a latent confounder between $X_i$ and $X_j$. We prove the contrapositive statement instead, i.e., if there is no latent confounder between $X_i$ and $X_j$, then $P(x_j \mid do(x_i), do(\mathbf{W} = \mathbf{w})) = P(x_j \mid x_i, do(\mathbf{W} = \mathbf{w}))$, $\forall x_i, x_j \in [K]$. Note that by construction, we have: $(\mathsf{Pa}(X_i) \cup \mathsf{Pa}(X_j) \setminus \{X_i\}) \subseteq \mathbf{W}$. For such choice of set $\mathbf{W}$ and the fact that $X_j \notin \mathsf{An}(X_i)$ and there is no latent confounder between $X_i$ and $X_j$, we have $(X_j \perp\!\!\!\perp X_i)_{\mathcal{G}_{X_i \overline{\mathbf{W}}}}$. Thus, from Pearl's do-calculus Rule 2, we have $P(x_j \mid do(x_i), do(\mathbf{W} = \mathbf{w})) = P(x_j \mid x_i, do(\mathbf{W} = \mathbf{w}))$, $\forall x_i, x_j \in [K]$. This completes the proof of the reverse direction.

### A.8 Proof of Lemma 4.3:

Suppose that Assumption 4.1 holds and we have access to $max(\frac{8}{\epsilon^2}, \frac{8}{\gamma^2}) \log \frac{2K^2}{\delta_1}$ samples from $do(X_i = x_i, \mathbf{W} = \mathbf{w}) \, \forall x_i \in [K]$ and $\frac{8}{\epsilon^2} \log \frac{2K^2}{\delta_2}$ samples from $do(\mathbf{W} = \mathbf{w})$ for a fixed $w \in [K]^{|\mathbf{W}|}$ for some $\mathbf{W} \subseteq \mathbf{V}$. We want to show that with probability at least $1 - \delta_1 - \delta_2$, we have the following:

$$(X_i \in \mathsf{An}(X_j))_{\mathcal{G}_{\overline{\mathbf{w}}}} \iff \exists \, x_i, x_j \in [K] \, s.t. \, \left| \widehat{P}(x_j|do(\mathbf{w})) - \widehat{P}(x_j|do(\mathbf{w}), do(x_i)) \right| > \frac{\epsilon}{2}. \quad (13)$$

Using Hoeffding's inequality with $A$ samples from intervention $do(x_i, \mathbf{w})$,

$$\left| \widehat{P}(x_j|do(x_i), do(\mathbf{w})) - P(x_j|do(x_i), do(\mathbf{w})) \right| \geq \sqrt{\frac{1}{2A} \log \frac{2K^2}{\delta_1}} \quad w.p. \ at \ most \ \frac{\delta_1}{K^2}. \quad (14)$$

If we choose $A = max(\frac{8}{\epsilon^2}, \frac{8}{\gamma^2}) \log \frac{2K^2}{\delta_1}$, we have:

$$\left| \widehat{P}(x_j|do(x_i), do(\mathbf{w})) - P(x_j|do(x_i), do(\mathbf{w})) \right| \geq \frac{\epsilon}{4} \quad w.p. \ at \ most \ \frac{\delta_1}{K^2}. \quad (15)$$

Similarly, using Hoeffding's inequality with $B$ samples from intervention $do(\mathbf{w})$,

$$\left| \widehat{P}(x_j|do(\mathbf{w})) - P(x_j|do(\mathbf{w})) \right| \geq \sqrt{\frac{1}{2A} \log \frac{2K^2}{\delta_1}} \quad w.p. \ at \ most \ \frac{\delta_2}{K^2}. \quad (16)$$

If we choose $B = \frac{8}{\epsilon^2} \log \frac{2K^2}{\delta_2}$, we have:

$$\left| \widehat{P}(x_j|do(\mathbf{w})) - P(x_j|do(\mathbf{w})) \right| \geq \frac{\epsilon}{4} \quad w.p. \ at \ most \ \frac{\delta_2}{K^2}. \quad (17)$$

Since the realization $\mathbf{w} \in [K]^{|\mathbf{W}|}$ is fixed, while $x_i$ and $x_j$ are in $[K]$, we have a total of $K^2$ possible bad events when the estimates are not accurate. Given the choice of samples, $A$ and $B$, we have:

$$\left| \widehat{P}(x_j|do(x_i), do(\mathbf{w})) - P(x_j|do(x_i), do(\mathbf{w})) \right| \leq \frac{\epsilon}{4} \quad \forall x_i, x_j \in [K] \ w.p. \ at \ least \ 1 - \delta_1, \quad (18)$$

$$\left| \widehat{P}(x_j|do(\mathbf{w})) - P(x_j|do(\mathbf{w})) \right| \leq \frac{\epsilon}{4} \quad \forall x_j \in [K] \ w.p. \ at \ least \ 1 - \delta_2. \quad (19)$$

Under the good event, which occurs with a probability of at least $1 - \delta_1 - \delta_2$, the estimates are accurate. We now consider the two possible scenarios. Suppose that $X_i \notin \mathsf{An}(X_j)$ in $\mathcal{G}_{\overline{\mathbf{W}}}$. In this case by Pearl's do-calculus Rule 3 we have $\left| P(x_j|do(x_i), do(\mathbf{w})) - P(x_j|do(\mathbf{w})) \right| = 0$ , $\forall x_i, x_j \in [K]$. By triangular inequality we have the following:

$$\left| \widehat{P}(x_j|do(x_i), do(\mathbf{w})) - \widehat{P}(x_j|do(\mathbf{w})) \right| \leq \left| \widehat{P}(x_j|do(x_i), do(\mathbf{w})) - P(x_j|do(x_i), do(\mathbf{w})) \right| +$$
$$\left| \widehat{P}(x_j|do(\mathbf{w})) - P(x_j|do(\mathbf{w})) \right| \leq \frac{\epsilon}{2} \quad \forall x_i, x_j \in [K]. \quad (20)$$

However, when $X_i \in \mathsf{An}(X_j)$ in $\mathcal{G}_{\overline{\mathbf{W}}}$ under Assumption 4.1 we must have some configuration say $x_i, x_j \in [K]$ for any $\mathbf{w} \in [K]^{|\mathbf{W}|}$ such that $\left| P(x_j|do(x_i), do(\mathbf{w})) - P(x_j|do(\mathbf{w})) \right| > \epsilon$. By triangular inequality when $X_i \in \mathsf{An}(X_j)$ in $\mathcal{G}_{\overline{\mathbf{W}}}$, $\exists \ x_i, x_j \in [K]$ such that

$$\left| \widehat{P}(x_j|do(x_i), do(\mathbf{w})) - \widehat{P}(x_j|do(\mathbf{w})) \right| \geq \left| P(x_j|do(x_i), do(\mathbf{w})) - P(x_j|do(\mathbf{w})) \right|$$
$$- \left| \widehat{P}(x_j|do(x_i), do(\mathbf{w})) - P(x_j|do(x_i), do(\mathbf{w})) \right| - \left| \widehat{P}(x_j|do(\mathbf{w})) - P(x_j|do(\mathbf{w})) \right| > \frac{\epsilon}{2}. \quad (21)$$

Thus, using Assumption 4.1 with the given choice of number of samples with probability at least $1 - \delta_1 - \delta_2$, we have the following result:

$$(X_i \in \mathsf{An}(X_j))_{\mathcal{G}_{\overline{\mathbf{W}}}} \iff \exists \ x_i, x_j \in [K] \ s.t. \ \left| \ \widehat{P}(x_j|do(\mathbf{w})) - \widehat{P}(x_j|do(\mathbf{w}), do(x_i)) \ \right| > \frac{\epsilon}{2}. \quad (22)$$

This completes the proof for Lemma 4.3.

### A.9 Proof of Lemma 4.4:

In order to prove that Algorithm 1 learns the true transitive closure under any intervention, i.e., $\mathcal{G}_{\overline{\mathbf{W}}}^{tc}$, we recall from the proof of Lemma 4.3 that the test for ancestrality works with high probability under the event that the causal effects of the form $P(x_j|do(x_i), do(\mathbf{w}))$ and $P(x_j|do(\mathbf{w}))$ are estimated accurately with an error of at most $\frac{\epsilon}{4}$ for all $x_i, x_j \in [K]$ and any fixed $\mathbf{w} \in [K]^{\mathbf{W}}$. Now, since Algorithm 1 takes $B = \frac{8}{\epsilon^2} \log \frac{2nK^2}{\delta_2}$ samples from $do(\mathbf{W} = \mathbf{w})$ and $A = \max\left(\frac{8}{\epsilon^2}, \frac{8}{\gamma^2}\right) \log \frac{2nK^2}{\delta_1}$ samples from every $do(X_i = x_i, \mathbf{W} = \mathbf{w})$ for all $X_i \in \mathbf{V} \setminus \mathbf{W}$ and for all $x_i \in [K]$, the total number of intervention samples collected is clearly at most $KAn + B$. In order to show that Algorithm 1 learns the true transitive closure under any intervention, i.e., $\mathcal{G}_{\overline{\mathbf{W}}}^{tc}$, with high probability, we must demonstrate that Algorithm 1 can estimate all causal effects with a maximum error of $\frac{\epsilon}{4}$ with high probability, so that all the ancestrality tests work with high probability, as implied by the proof of Lemma 4.3.

Using Hoeffding's inequality with $B = \frac{8}{\epsilon^2} \log \frac{2nK^2}{\delta_2}$ samples from the intervention $do(\mathbf{w})$, we have for any $X_j \in \mathbf{V} \setminus \mathbf{W}$:

$$\left| \widehat{P}(x_j | do(\mathbf{w})) - P(x_j | do(\mathbf{w})) \right| \leq \frac{\epsilon}{4} \quad \forall x_j \in [K] \quad w.p. \text{ at least } 1 - \frac{\delta_2}{n}. \tag{23}$$

Using the union bound we have the following:

$$\left| \widehat{P}(X_j = x_j | do(\mathbf{w})) - P(X_j = x_j | do(\mathbf{w})) \right| \leq \frac{\epsilon}{4} \quad \forall x_j \in [K] , \; \forall X_j \in \mathbf{V} \setminus \mathbf{W} \quad w.p. \text{ at least } 1 - \delta_2. \tag{24}$$

Now, consider a fixed pair $X_i, X_j \in \mathbf{V} \setminus \mathbf{W}$, and using $A = \max\left(\frac{8}{\epsilon^2}, \frac{8}{\gamma^2}\right) \log \frac{2nK^2}{\delta_1}$ samples from the intervention $do(x_i, \mathbf{w})$ for every $x_i \in [K]$, we have the following using Hoeffding's inequality:

$$\left| \widehat{P}(x_j | do(x_i), do(\mathbf{w})) - P(x_j | do(x_i), do(\mathbf{w})) \right| \leq \min(\frac{\epsilon}{4}, \frac{\gamma}{4}) \quad \forall x_i, x_j \in [K] \quad w.p. \text{ at least } 1 - \frac{\delta_1}{n} \tag{25}$$

Using the union bound we have the following:

$$\left| \widehat{P}(x_j | do(x_i), do(\mathbf{w})) - P(x_j | do(x_i), do(\mathbf{w})) \right| \leq \min(\frac{\epsilon}{4}, \frac{\gamma}{4})$$
$$\forall x_i, x_j \in [K] , \; \forall X_j \in \mathbf{V} \setminus (\mathbf{W} \cup \{X_i\}) \quad w.p. \text{ at least } 1 - \delta_1 \tag{26}$$

Again using the union bound over all intervention targets $X_i \in \mathbf{V}$ we have the following:

$$\left| \widehat{P}(x_j | do(x_i), do(\mathbf{w})) - P(x_j | do(x_i), do(\mathbf{w})) \right| \leq \min(\frac{\epsilon}{4}, \frac{\gamma}{4})$$
$$\forall x_i, x_j \in [K] , \; \forall X_i \in \mathbf{V} \setminus \mathbf{W} , \; \forall X_j \in \mathbf{V} \setminus (\mathbf{W} \cup \{X_i\}) \quad w.p. \text{ at least } 1 - n\delta_1 \tag{27}$$

From Equations 24 and 27, using the union bound with probability at least $1 - n\delta_1 - \delta_2$, all the causal effects are estimated within an error of $\frac{\epsilon}{4}$ from the true values, ensuring that all ancestrality tests work perfectly under this good event. Thus, Algorithm 1 learns the true transitive closure under any intervention, i.e., $\mathcal{G}^{tc}_{\overline{\mathbf{W}}}$, with $KAn + B$ intervention samples with probability of at least $1 - n\delta_1 - \delta_2$. Also, if we set $\delta_1 = \frac{\delta}{2n}$ and $\delta_2 = \frac{\delta}{2}$, then Algorithm 1 learns the true transitive closure under any intervention, i.e., $\mathcal{G}^{tc}_{\overline{\mathbf{W}}}$ with a probability of $1 - \delta$, with $KAn + B$ intervention samples, where $A = \max\left(\frac{8}{\epsilon^2}, \frac{8}{\gamma^2}\right) \log \frac{4n^2K^2}{\delta}$ and $B = \frac{8}{\epsilon^2} \log \frac{4nK^2}{\delta}$. This completes the proof of Lemma 4.4.

### A.10   Proof of Theorem 4.1:

We start by revising the statement of Lemma 4.3: Algorithm 1 learns the true transitive closure under any intervention, i.e., $\mathcal{G}^{tc}_{\overline{\mathbf{W}}}$, with $KAn + B$ intervention samples with a probability of at least $1 - n\delta_1 - \delta_2$. Algorithm 2 randomly samples a target set $\mathbf{W}$ and calls Algorithm 1 to learn the active true transitive closure of the post-interventional graph, i.e., $\mathcal{G}^{tc}_{\overline{\mathbf{W}}}$. For every iteration, Algorithm 2 computes transitive reduction $Tr(\mathcal{G}^{tc}_{\overline{\mathbf{W}}})$ and updates all the edges to construct the observable graph. To prove the results in Theorem 4.1, we rely on Lemma 5 from [14], which is stated below:

**Lemma A.2.** *[14] Consider a graph $\mathcal{G}$ with observed variables $V$ and an intervention set $\mathbf{W} \subseteq V$. Consider post-interventional observable graph $\mathcal{G}_{\overline{\mathbf{W}}}$ and a variable $X_j \in V \setminus \mathbf{W}$. Let $X_i \in \mathsf{Pa}(X_j)$ be such that all the parents of $X_j$ above $X_i$ in partial order are included in the intervention set $\mathbf{W}$. This implies that $\{W_i : \pi(W_i) > \pi(X_i) \ \& \ W_i \in \mathsf{Pa}(X_j)\} \subseteq \mathbf{W}$. Then, the directed edge $(X_i, X_j) \in \mathbf{E}(Tr(\mathcal{G}_{\overline{\mathbf{W}}}))$. The properties of transitive reduction yields $Tr(\mathcal{G}_{\overline{\mathbf{W}}}) = Tr(\mathcal{G}_{\overline{\mathbf{W}}}^{tc})$. Consequently, the transitive reduction of $\mathcal{G}_{\overline{\mathbf{W}}}^{tc}$, i.e., $Tr(\mathcal{G}_{\overline{\mathbf{W}}}^{tc}) = Tr(\mathcal{G}_{\overline{\mathbf{W}}})$ may be used to learn the directed edge $(X_i, X_j)$.*
*(Note: $\mathbf{E}(\mathcal{G})$ denotes the edges of the graph $\mathcal{G}$ and $\pi$ is any total order that is consistent with the partial order implied by the DAG, i.e., $\pi(X) < \pi(Y)$ iff X is an ancestor of Y).*

Assume that the number of the direct parents of $X_j$ above $X_i$ is $d_{ij}$ where $d_{ij} \leq d_{max}$. Let $\mathcal{E}_i(X_j)$ be the following event: $X_i, X_j \notin \mathbf{W} \ \& \ \{W_i : \pi(W_i) > \pi(X_i) \ \& \ W_i \in \mathsf{Pa}(X_j)\} \subseteq \mathbf{W}$. The probability of this event for one run of the outer loop in Algorithm 2 with the assumption that $2d_{max} >= 2$ is given by:

$$P[\mathcal{E}_i(X_j)] = \frac{1}{4d_{max}^2}(1 - \frac{1}{2d_{max}})^{d_{ij}} \geq \frac{1}{4d_{max}^2}(1 - \frac{1}{2d_{max}})^{2d_{max}} \geq \frac{1}{d_{max}^2}\frac{1}{16}. \quad (28)$$

The last inequality holds for $2d_{max} >= 2$ because $(1 - \frac{1}{x})^x \geq 0.25, \ \forall x \geq 2$. Based on Lemma A.2, the event $\mathcal{E}_i(X_j)$ implies that the directed edge $(X_j, X_j)$ will be present in $Tr(\mathcal{G}_{\overline{\mathbf{W}}}^{tc})$ and will be learned. The outer loop runs for $8\alpha d_{max} \log(n)$ iterations and elements of the set $\mathbf{W}$ are independently sampled. The probability of failure, i.e., the event under consideration does not happen for all runs of the outer loop in Algorithm 2, is bounded as follows:

$$P[(\mathcal{E}_i(V))^c] \leq (1 - \frac{1}{16 \, d_{max}^2})^{8\alpha d_{max} \log(n)} \leq e^{-\frac{\alpha}{2d_{max}} \log(n)} = \frac{1}{n^{\frac{\alpha}{2d_{max}}}}. \quad (29)$$

For a graph with a total number of variables $n$, the total number of such bad events will be $\binom{n}{2}$ since a graph can have at most $\binom{n}{2}$ edges. Using the union bound, the probability of bad event for any pair of variables is given by:

$$P[Failure] \leq \binom{n}{2} \times \frac{1}{n^{\frac{\alpha}{2d_{max}}}} \leq \frac{1}{n^{\frac{\alpha}{2d_{max}} - 2}}. \quad (30)$$

Under the event that Algorithm 1 learns the correct transitive closure $\mathcal{G}_{\overline{\mathbf{W}}}^{tc}$ for all the $8\alpha d_{\max} \log n$ randomly sampled intervention sets $\mathbf{W} \subseteq \mathbf{V}$, the above derivation shows that we will be able to learn all edges in the true observable graph with a probability of at least $1 - \frac{1}{n^{\frac{\alpha}{2d_{\max}} - 2}}$. Now recall the result from Lemma 4.3 that Algorithm 1 learns the true transitive closure under any intervention, i.e., $\mathcal{G}_{\overline{\mathbf{W}}}^{tc}$, with $KAn + B$ intervention samples with a probability of at least $1 - n\delta_1 - \delta_2$. Combining the two results above using the union bound, we have the following result:

Algorithm 2 learns the true observable graph with a probability of at least $1 - \frac{1}{n^{\frac{\alpha}{2d_{\max}} - 2}} - 8\alpha d_{\max} \log(n)(n\delta_1 + \delta_2)$ with a maximum $8\alpha d_{\max} \log n(KAn + B)$ interventional samples. Also, if we set $\alpha = \frac{2d_{\max} \log(\frac{2}{\delta} + 2)}{\log n}$, $\delta_1 = \frac{\delta}{32\alpha d_{\max} n \log n}$, and $\delta_2 = \frac{\delta}{32\alpha d_{\max} \log n}$, then Algorithm 2 learns the true observable graph with a probability of at least $1 - \delta$. Where $A = \max\left(\frac{8}{\epsilon^2}, \frac{8}{\gamma^2}\right) \log \frac{2nK^2}{\delta_1}$ and $B = \frac{8}{\epsilon^2} \log \frac{2nK^2}{\delta_2}$. This completes the proof of Theorem 4.1.

## A.11 Proof of Lemma 4.5:

Consider two nodes $X_i$ and $X_j$ s.t. $X_j \notin \mathsf{An}(X_i)$ and suppose that Assumptions 4.2 4.3 holds and we have access to $max(\frac{8}{\epsilon^2}, \frac{8}{\gamma^2}) \log \frac{2K^2}{\delta_1}$ samples from $do(X_i = x_i, \mathbf{W} = \mathbf{w}) \ \forall x_i \in [K]$ and $\frac{16}{\eta\gamma^2} \log(\frac{2K^2}{\delta_3}) + \frac{1}{2\eta^2} \log(\frac{2K^2}{\delta_4})$ from $do(\mathbf{W} = \mathbf{w})$ for a fixed $w \in [K]^{|\mathbf{W}|}$ and $\mathbf{W} \subseteq \mathbf{V}$ such that $(\mathsf{Pa}(X_i) \cup \mathsf{Pa}(X_j) \setminus \{X_i\}) \subseteq \mathbf{W}$ and $X_i \ \& \ X_j \notin \mathbf{W}$. We want to show that, with probability at least $1 - \delta_1 - \delta_3 - \delta_4$, we have the following:

There exists a latent confounder between $X_i$ and $X_j \iff$

$$\exists\, x_i, x_j \in [K] \ s.t. \ \big| \ \widehat{P}(x_j|do(x_i), do(\mathbf{w})) - \widehat{P}(x_j|x_i, do(\mathbf{w})) \ \big| > \frac{\gamma}{2}. \tag{31}$$

Using Hoeffding's inequality with $A$ samples from intervention $do(x_i, \mathbf{w})$.

$$\left| \widehat{P}(x_j|do(x_i), do(\mathbf{w})) - P(x_j|do(x_i), do(\mathbf{w})) \right| \geq \sqrt{\frac{1}{2A} \log \frac{2K^2}{\delta_1}} \ \ w.p. \ at \ most \ \frac{\delta_1}{K^2}. \tag{32}$$

If we choose $A = max(\frac{8}{\epsilon^2}, \frac{8}{\gamma^2}) \log \frac{2K^2}{\delta_1}$, we have:

$$\left| \widehat{P}(x_j|do(x_i), do(\mathbf{w})) - P(x_j|do(x_i), do(\mathbf{w})) \right| \geq \frac{\gamma}{4} \ \ w.p. \ at \ most \ \frac{\delta_1}{K^2}. \tag{33}$$

Using Hoeffding's inequality with $C$ samples from intervention $do(x_i)$.

$$\left| \widehat{P}(x_j|x_i, do(\mathbf{w})) - P(x_j|x_i, do(\mathbf{w})) \right| \geq \sqrt{\frac{1}{2C_{x_i}} \log \frac{2K^2}{\delta_3}} \ \ w.p. \ at \ most \ \frac{\delta_3}{K^2}. \tag{34}$$

Where $C_{x_i}$ is the number of samples where $X_i = x_i$ among the $C$ samples for the intervention $do(\mathbf{w})$. Note the we can't directly control $C_{x_i}$ and it's value depends on the true interventions distribution $P(x_i, do(\mathbf{w}))$ along-with the number of samples $C$. Suppose if we can set $C_{x_i} \geq \frac{8}{\gamma^2} \log \frac{2K^2}{\delta_3}$, we have:

$$\left| \widehat{P}(x_j|x_i, do(\mathbf{w})) - P(x_j|x_i, do(\mathbf{w})) \right| \geq \frac{\gamma}{4} \ \ w.p. \ at \ most \ \frac{\delta_3}{K^2}. \tag{35}$$

We need to find the number of samples $C$ such that $C_{x_i} \geq \frac{8}{\gamma^2} \log \frac{2K^2}{\delta_3}$. Using the Hoeffding's bound we have:

$$P(C_{x_i} \geq CP(x_i|do(\mathbf{w})) - \eta) \geq 1 - 2e^{-2\eta^2/C}. \tag{36}$$

Let $\frac{\delta_4}{K^2} = 2e^{-2\eta^2/C}$, which implies $\eta = \sqrt{\frac{C}{2} \log \frac{2K^2}{\delta_4}}$. Thus we have:

$$P\left( C_{x_i} \geq CP(x_i|do(\mathbf{w})) - \sqrt{\frac{C}{2} \log \frac{2K^2}{\delta_4}} \right) \geq 1 - \frac{\delta_4}{K^2} \tag{37}$$

$$C_{x_i} \geq CP(x_i|do(\mathbf{w})) - \sqrt{\frac{C}{2} \log \frac{2K^2}{\delta_4}} \ \ w.p. \ at \ least \ 1 - \frac{\delta_4}{K^2}. \tag{38}$$

Using Assumption 4.3, we have $P(x_i|do(\mathbf{w})) = 0$ or $P(x_i|do(\mathbf{w})) \geq \eta$. Note that if $P(x_i|do(\mathbf{w})) = 0$, the event will never happen, and we don't care about the accuracy of the estimate $\widehat{P}(x_j|x_i, do(\mathsf{w}))$ because it is already initialized to zero. Now the equation above can be rewritten as:

$$C_{x_i} \geq C\eta - \sqrt{\frac{C}{2} \log \frac{2K^2}{\delta_4}} \ \ w.p. \ at \ least \ 1 - \frac{\delta_4}{K^2}. \tag{39}$$

Since we want $C_{x_i} \geq \frac{8}{\gamma^2} \log \frac{2K^2}{\delta_3}$ with high probability, we have the following relationship:

$$C\eta - \sqrt{\frac{C}{2}\log\frac{2K^2}{\delta_4}} \geq \frac{8}{\gamma^2}\log\frac{2K^2}{\delta_3} \tag{40}$$

Solving the equation for number of samples $C$ we get:

$$C \geq \frac{4\eta\frac{8\log\left(\frac{2K^2}{\delta_3}\right)}{\gamma^2} + \ln\left(\frac{2K^2}{\delta_4}\right) + \sqrt{8\eta\frac{8\log\left(\frac{2K^2}{\delta_3}\right)}{\gamma^2}\ln\left(\frac{2K^2}{\delta_4}\right) + \ln^2\left(\frac{2K^2}{\delta_4}\right)}}{4\eta^2} \tag{41}$$

In order to make the expression simpler we choose the number of samples $C$ as follows:

$$C = \frac{4\eta\frac{8\log\left(\frac{2K^2}{\delta_3}\right)}{\gamma^2} + \ln\left(\frac{2K^2}{\delta_4}\right) + \sqrt{8\eta\frac{8\log\left(\frac{2K^2}{\delta_3}\right)}{\gamma^2}\ln\left(\frac{2K^2}{\delta_4}\right) + \ln^2\left(\frac{2K^2}{\delta_4}\right) + (4\eta\frac{8\log\left(\frac{2K^2}{\delta_3}\right)}{\gamma^2})^2}}{4\eta^2} \tag{42}$$

$$C = \frac{4\eta\frac{8\log\left(\frac{2K^2}{\delta_3}\right)}{\gamma^2} + \ln\left(\frac{2K^2}{\delta_4}\right) + \sqrt{\left(4\eta\frac{8\log\left(\frac{2K^2}{\delta_3}\right)}{\gamma^2} + \ln\left(\frac{2K^2}{\delta_4}\right)\right)^2}}{4\eta^2} \tag{43}$$

$$C = \frac{4\eta\frac{8\log\left(\frac{2K^2}{\delta_3}\right)}{\gamma^2} + \ln\left(\frac{2K^2}{\delta_4}\right)}{2\eta^2} \tag{44}$$

$$C = \frac{16}{\eta\gamma^2}\log(\frac{2K^2}{\delta_3}) + \frac{1}{2\eta^2}\log(\frac{2K^2}{\delta_4}) \tag{45}$$

Suppose we take $C$ samples for intervention $do(\mathbf{w})$ as given above. Now, from Equations 35, 39, and 40, using the union bound, we have the following:

$$\left|\widehat{P}(x_j|x_i, do(\mathbf{w})) - P(x_j|x_i, do(\mathbf{w}))\right| \geq \frac{\gamma}{4} \;\; w.p. \; at \; most \; \frac{\delta_3 + \delta_4}{K^2}. \tag{46}$$

Since the realization $\mathbf{w} \in [K]^{|\mathbf{W}|}$ is fixed, but $x_i, x_j \in [K]$, we have a total of $K^2$ possible bad events when estimates are not good. With the given choice of number of samples $A$ and $C$, we have:

$$\left|\widehat{P}(x_j|do(x_i), do(\mathbf{w})) - P(x_j|do(x_i), do(\mathbf{w}))\right| \leq \frac{\gamma}{4} \;\; \forall x_j \in [K] \;\; w.p. \; at \; least \; 1 - \delta_1. \tag{47}$$

$$\left|\widehat{P}(x_j|x_i, do(\mathbf{w})) - P(x_j|x_i, do(\mathbf{w}))\right| \leq \frac{\gamma}{4} \;\; \forall x_i, x_j \in [K] \;\; w.p. \; at \; least \; 1 - \delta_3 - \delta_4. \tag{48}$$

Under the good event, which has a probability of at least $1 - \delta_1 - \delta_3 - \delta_4$, both estimates are accurate. We now consider the two possible scenarios. Suppose that there is no latent confounder between $X_i$ and $X_j$. In this case by Lemma 4.2 we have $\left|P(x_j|do(x_i), do(\mathbf{w})) - P(x_j|x_i, do(\mathbf{w}))\right| = 0$, $\forall x_i x_j \in [K]$. By triangular inequality we have the following:

$$\left|\widehat{P}(x_j|do(x_i), do(\mathbf{w})) - \widehat{P}(x_j|x_i, do(\mathbf{w}))\right| \leq \left|\widehat{P}(x_j|do(x_i), do(\mathbf{w})) - P(x_j|do(x_i), do(\mathbf{w}))\right| +$$

$$\left|\widehat{P}(x_j|x, do(\mathbf{w})) - P(x_j|x_i, do(\mathbf{w}))\right| \leq \frac{\gamma}{2} \;\; \forall x_i, x_j \in [K]. \tag{49}$$

However, when there is a latent confounder between $X_i$ and $X_j$, in this case, under Assumption 4.2, we must have some configuration, say $x_i, x_j \in [K]$, for any $\mathbf{w} \in [K]^{|\mathbf{W}|}$, such that $\left| P(x_j|do(x_i), do(\mathbf{w})) - P(x_j|x_i, do(\mathbf{w})) \right| > \gamma$. By triangular inequality when there is a latent confounder between $X_i$ and $X_j$, $\exists\, x_i, x_j \in [K]$ such that:

$$
\left| \widehat{P}(x_j|do(x_i), do(\mathbf{w})) - \widehat{P}(x_j|x_i, do(\mathbf{w})) \right| \geq \left| P(x_j|do(x_i), do(\mathbf{w})) - P(x_j|x_i, do(\mathbf{w})) \right|
$$
$$
- \left| \widehat{P}(x_j|do(x_i), do(\mathbf{w})) - P(x_j|do(x_i), do(\mathbf{w})) \right| - \left| \widehat{P}(x_j|x_i, do(\mathbf{w})) - P(x_j|x_i, do(\mathbf{w})) \right| > \frac{\gamma}{2}
$$
(50)

Thus, using Assumption 4.2 with the given choice of number of samples with probability at least $1 - \delta_1 - \delta_3 - \delta_4$, we have the following result:

There exists a latent confounder between $X_i$ and $X_j$ $\iff$
$$
\exists\, x_i, x_j \in [K] \ s.t. \ \left| \ \widehat{P}(x_j|do(x_i), do(\mathbf{w})) - \widehat{P}(x_j|x_i, do(\mathbf{w})) \ \right| > \frac{\gamma}{2}.
$$
(51)

This completes the proof for Lemma 4.5.

### A.12 Proof of Theorem 4.2:

The Algorithm 3 first calls Algorithm 2 to learn the observable graph structure. We have already proved in Theorem 4.1 that Algorithm 2 learns the true observable graph with a probability of at least $1 - \frac{1}{n^{\frac{\alpha}{2d_{\max}} - 2}} - 8\alpha d_{\max} \log(n)(n\delta_1 + \delta_2)$ with a maximum of $8\alpha d_{\max} \log n(KAn + B)$ interventional samples. The next phase in Algorithm 3 is to learn/detect latent confounders between any pair of variables. For all pairs of nodes $X_i$ and $X_j$ such that $X_j \notin \mathsf{An}(X_i)$, we define a set of nodes $\mathbf{W}_{ij} \subseteq \mathbf{V}$ such that $X_i, X_j \notin \mathbf{S}_i$, where $\mathbf{W}_{ij} = (\mathsf{Pa}(X_i) \cup \mathsf{Pa}(X_j) \setminus \{X_i\})$. Also, note that $|\mathbf{W}_{ij}| \leq 2d_{\max}$. Let us define the event $\mathcal{E}_{ij} = [\mathbf{W}_{ij} \subseteq \mathbf{W} \ \& \ X_j, X_i \notin \mathbf{W}]$. The probability of this event for one run of the outer loop in Algorithm 2 with the assumption that $2d_{\max} \geq 2$ is given by:

$$
P[\mathcal{E}_{ij}] = \frac{1}{4d_{max}^2}\left(1 - \frac{1}{2d_{max}}\right)^{|\mathbf{W}_{ij}|} \geq \frac{1}{4d_{max}^2}\left(1 - \frac{1}{2d_{max}}\right)^{2d_{max}} \geq \frac{1}{d_{max}^2}\frac{1}{16}.
$$
(52)

The last inequality holds for $d_{\max} \geq 2$. Note that we reuse all the interventional data samples from Algorithm 2 in Algorithm 3. Under Assumption 4.2, if the event $\mathcal{E}_{ij}$ happens with a large enough number of samples, we can detect the presence or absence of latent confounders between $X_i$ and $X_j$. The outer loop runs for $8\alpha d_{max} \log(n)$ iterations, and the elements of the set $\mathbf{W}$ are independently sampled. The probability of failure, i.e., the event under consideration does not happen for all runs of the outer loop in Algorithm 2, is bounded as follows:

$$
P[\mathcal{E}_{ij}^c] \leq \left(1 - \frac{1}{16\,d_{max}^2}\right)^{8\alpha d_{max} \log(n)} \leq e^{-\frac{\alpha}{2d_{max}}\log(n)} = \frac{1}{n^{\frac{\alpha}{2d_{max}}}}.
$$
(53)

For a graph with a total number of variables $n$, the total number of such bad events will be $\binom{n}{2}$. Using the union bound, the probability of bad event for any pair of variables is given by:

$$
P[Failure] \leq \binom{n}{2} \times \frac{1}{n^{\frac{\alpha}{2d_{max}}}} \leq \frac{1}{n^{\frac{\alpha}{2d_{max}} - 2}}.
$$
(54)

This implies with a probability of $1 - \frac{1}{n^{\frac{\alpha}{2d_{\max}} - 2}}$, we will be able to find an appropriate interventional dataset to test the presence of latent confounders between any pair of variables using Assumption 4.2

after running Algorithm 2. We still need to make sure we have enough interventional samples to be able to test the latents. This is because we need to accurately estimate conditional effects to carry out the test, as in Assumption 4.2. We first consider estimation of the causal effect $\widehat{P}(x_j|do(x_i), do(\mathbf{w}))$ for any randomly sampled set $\mathbf{W}$. Now, consider a fixed $X_i, X_j \in \mathbf{V} \setminus \mathbf{W}$. We have access to $\max\left(\frac{8}{\epsilon^2}, \frac{8}{\gamma^2}\right) \log \frac{2nK^2}{\delta_1}$ samples for every $x_i \in [K]$. We have already shown that under the good event, we have the following:

$$\left|\widehat{P}(x_j|do(x_i), do(\mathbf{w})) - P(x_j|do(x_i), do(\mathbf{w}))\right| \leq \min(\frac{\epsilon}{4}, \frac{\gamma}{4})$$

$$\forall x_i, x_j \in [K], \ \forall X_i \in \mathbf{V} \setminus \mathbf{W}, \ \forall X_j \in \mathbf{V} \setminus (\mathbf{W} \cup \{X_i\}) \ w.p. \ at \ least \ 1 - n\delta_1 \tag{55}$$

Now, we consider estimation of the conditional causal effects, i.e., $\widehat{P}(x_j|x_i, do(\mathbf{w}))$. Note the while running the Algorithm 2 we have access to $B = \frac{8}{\epsilon^2} \log \frac{2nK^2}{\delta_2}$ samples form intervention $do(\mathbf{w})$ and in the step 7 of Algorithm 3 we add more samples to the data set and have access to at least $C = \frac{16}{\eta\gamma^2} \log(\frac{2n^2K^2}{\delta_3}) + \frac{1}{2\eta^2} \log(\frac{2n^2K^2}{\delta_4})$ samples instead. Now, consider a fixed $X_i, X_j \in \mathbf{V} \setminus \mathbf{W}$. With access to $C$ samples as given above, following from Equation 48 in the Proof of Lemma 4.5, we have the following result:

$$\left|\widehat{P}(x_j|x_i, do(\mathbf{w})) - P(x_j|x_i, do(\mathbf{w}))\right| \leq \frac{\gamma}{4} \ \forall x_i, x_j \in [K] \ w.p. \ at \ least \ 1 - \frac{\delta_3}{n^2} - \frac{\delta_4}{n^2}. \tag{56}$$

Note that in the above equation, we have $\frac{\delta_3}{n^2}$ and $\frac{\delta_4}{n^2}$ instead of $\delta_3$ and $\delta_4$ as in Equation 48, because here in the number of samples $C$, we also have $\frac{\delta_3}{n^2}$ and $\frac{\delta_4}{n^2}$ instead of $\delta_3$ and $\delta_4$ when compared to the number of samples in Equation 45. Now, using the union bound we have the following:

$$\left|\widehat{P}(x_j|x_i, do(\mathbf{w})) - P(x_j|x_i, do(\mathbf{w}))\right| \leq \frac{\gamma}{4}$$

$$\forall x_i, x_j \in [K], \ \forall X_j \in \mathbf{V} \setminus (\mathbf{W} \cup \{X_i\}) \ w.p. \ at \ least \ 1 - \frac{\delta_3}{n} - \frac{\delta_4}{n}. \tag{57}$$

Again using the union bound over all $X_i \in \mathbf{V} \setminus \mathbf{W}$ we have the following:

$$\left|\widehat{P}(x_j|do(x_i), do(\mathbf{w})) - P(x_j|do(x_i), do(\mathbf{w}))\right| \leq \frac{\gamma}{4}$$

$$\forall x_i, x_j \in [K], \ \forall X_i \in \mathbf{V} \setminus \mathbf{W}, \ \forall X_j \in \mathbf{V} \setminus (\mathbf{W} \cup \{X_i\}) \ w.p. \ at \ least \ 1 - \delta_3 - \delta_4 \tag{58}$$

This implies that under the good event, for every randomly sampled intervention set $\mathbf{W} \subseteq \mathbf{V}$, the estimate of the conditional causal effect is accurate within the desired $\frac{\gamma}{4}$ threshold. This would imply that the test for detection of latent variables is perfect under this good event. We have already shown that to ensure we have access to sufficient datasets to detect latent variables between any pair of nodes, the $8\alpha d_{\max} \log n$ randomly sampled target sets in Algorithm 2 are sufficient. Combining these results with the results from Theorem 4.1, we have the following:

The Algorithm 3 learns the true causal graph along with all latents with a probability of at least $1 - \frac{1}{n^{\frac{\alpha}{2d_{max}} - 2}} - \frac{1}{n^{\frac{\alpha}{2d_{max}} - 2}} - 8\alpha d_{max} \log(n)(n\delta_1 + \delta_2) - 8\alpha d_{max} \log(n)(\delta_3 + \delta_4) = 1 - \frac{2}{n^{\frac{\alpha}{2d_{max}} - 2}} - 8\alpha d_{max} \log(n)(n\delta_1 + (\delta_2 + \delta_3 + \delta_4))$ with a maximum $8\alpha d_{max} \log n(KAn + \max(B, C))$ interventional samples. Also If we set $\alpha = \frac{2d_{max} \log(\frac{4}{\delta} + 2)}{\log n}$, $\delta_1 = \frac{\delta}{64\alpha d_{max} n \log n}$ and $\delta_2 = \delta_3 = \delta_4 = \frac{\delta}{64\alpha d_{max} \log n}$, then Algorithm 2 learns the true causal graph with latents with a probability at least $1 - \delta$. Note that: $A = \max\left(\frac{8}{\epsilon^2}, \frac{8}{\gamma^2}\right) \log \frac{2nK^2}{\delta_1}$, $B = \frac{8}{\epsilon^2} \log \frac{2nK^2}{\delta_2}$, $C = \frac{16}{\eta\gamma^2} \log(\frac{2K^2}{\delta_3}) + \frac{1}{2\eta^2} \log(\frac{2K^2}{\delta_4})$. This completes the proof for Theorem 4.2.

---

**Algorithm 6:** Full version of Algorithm for causal bandits with unknown graph structure

---

**1** Set the Parameter $\delta, d_{max}$

**2** Calculate $\alpha, \delta_1, \delta_2, \delta_3, \delta_4$ as in Theorem 5.1

**3** $\mathcal{G}^{tc} = \mathsf{LearnTransitiveClosure}(\mathbf{W} = \phi, \frac{\delta}{2n}, \frac{\delta}{n})$

**4** $\mathcal{G}, \mathcal{IData} = \mathsf{LearnObservableGraph}(\mathsf{An}(Y)_{\mathcal{G}^{tc}}, \alpha, d_{max}, \delta_1, \delta_2)$

**5** $C = \frac{16}{\eta\gamma^2} \log(\frac{2n^2K^2}{\delta_3}) + \frac{1}{2\eta^2} \log(\frac{2n^2K^2}{\delta_4})$ , $B = \frac{8}{\epsilon^2} \log \frac{2nK^2}{\delta_2}$

**6** #Learn the bi-directed edges between reward $Y$ and all nodes $X_i \in \mathsf{An}(Y)$ and update $\mathcal{G}$.

**7** **for** *every* $X_i \in \mathsf{An}(Y)_{\mathcal{G}^{tc}}$ **do**

**8** $\quad$ Set $X_j := Y$

**9** $\quad$ Find interventional data sets $do(\mathbf{W} = \mathbf{w})$ and $do(X_i = x_i, \mathbf{W} = \mathbf{w})$ from $\mathcal{IData}$ s.t. $(\mathsf{Pa}(X_i) \cup \mathsf{Pa}(X_j) \setminus \{X_i\}) \subseteq \mathbf{W}$ and $X_i$ & $X_j \notin \mathbf{W}$

**10** $\quad$ Get $\max(0, B - C)$ new samples for $do(\mathbf{W} = \mathbf{w})$

**11** $\quad$ **if** $\exists\ x_i, x_j \in [K]\ s.t.\ |\widehat{P}(x_j|do(x_i), do(\mathbf{w})) - \widehat{P}(x_j|x_i, do(\mathbf{w}))| > \frac{\gamma}{2}$ **then**

**12** $\quad\quad$ Add bi-dirceted edge $X_i \leftrightarrow X_j$ to graph $\mathcal{G}$

**13** **while** *There is a new pair that is tested* **do**

**14** $\quad$ Find a new pair $(Z, X)$ s.t. $Z \in \mathsf{An}(Y)$ such that $Z$ and $Y$ don't have a bi-directed edge between them in $\mathcal{G}$ and $X \in \mathsf{MUCT}(\mathcal{G}_{\overline{\mathsf{Pa}(Z), \mathsf{Bi}(Z, \mathcal{G})}}, Y)$

**15** $\quad$ # Test for the latent between the pair $(Z, X)$ and update $\mathcal{G}$.

**16** $\quad$ Set $X_i := Z, X_j := X$

**17** $\quad$ if $X_j \in \mathsf{An}(X_i)$ swap them.

**18** $\quad$ Find interventional data sets $do(\mathbf{W} = \mathbf{w})$ and $do(X_i = x_i, \mathbf{W} = \mathbf{w})$ from $\mathcal{IData}$ s.t. $(\mathsf{Pa}(X_i) \cup \mathsf{Pa}(X_j) \setminus \{X_i\}) \subseteq \mathbf{W}$ and $X_i$ & $X_j \notin \mathbf{W}$

**19** $\quad$ Get $\max(0, B - C)$ new samples for $do(\mathbf{W} = \mathbf{w})$

**20** $\quad$ **if** $\exists\ x_i, x_j \in [K]\ s.t.\ |\widehat{P}(x_j|do(x_i), do(\mathbf{w})) - \widehat{P}(x_j|x_i, do(\mathbf{w}))| > \frac{\gamma}{2}$ **then**

**21** $\quad\quad$ Add bi-directed edge $X_i \leftrightarrow X_j$ to graph $\mathcal{G}$

**22** Learn the set of POMISs $\mathcal{I}_{\mathcal{G}}$ from the graph $\mathcal{G}$ Using Algorithm 1 in [5].

**23** Run UCB algorithm over the arm set $\mathcal{A} = \{\Omega(I) \mid \forall I \in \mathcal{I}_{\mathcal{G}}\}$.

### A.13 Full Version of Algorithm 4 and Proof of Theorem 5.1:

Algorithm 4 or its full version (Algorithm 6) starts by learning the transitive closure of the graph, denoted as $\mathcal{G}^{tc}$. This is because $\mathcal{G}^{tc}$ can give us $\mathsf{An}(Y)$, and every possible POMIS is a subset of $\mathsf{An}(Y)$. Thus, we can restrict ourselves to ancestors of the read node. From Lemma 4.4, we can learn the transitive closure $\mathcal{G}^{tc}$ with a probability of at least $1 - \delta$ with a maximum of $KAn + B$ interventional samples by setting $\delta_1 = \frac{\delta}{2n}$ and $\delta_2 = \frac{\delta}{2}$. Then, Algorithm 1 learns the true transitive closure with a probability of at least $1 - \delta$. (We have $A = \max\left(\frac{8}{\epsilon^2}, \frac{8}{\gamma^2}\right) \log \frac{2nK^2}{\delta_1}$ and $B = \frac{8}{\epsilon^2} \log \frac{2nK^2}{\delta_2}$ as in line 2 of Algorithm 1). Thus, the total interventional samples for this step turn out to be $Kn \max\left(\frac{8}{\epsilon^2}, \frac{8}{\gamma^2}\right) \log \frac{4n^2K^2}{\delta} + \frac{8}{\epsilon^2} \log \frac{4nK^2}{\delta}$.

The next step is to learn the complete observable graph induced on the reward node and its ancestors and then learn/detect only a subset of latent confounders which are characterized to be necessary and sufficient to learn the true set of POMISs (Theorem 3.1). Although this step saves us interventional samples compared to the full discovery Algorithm 3, which learns/detects latents between all pairs of variables, the exact saving will depend on the structure of the underlying causal graph. For the regret upper bound, we can use the results from Theorem 4.2 to bound the number of interventional samples for learning the true POMIS set from the ancestors of the reward node. This implies that given the true set of ancestors of the reward $\mathsf{An}(Y)$, we can learn the true POMIS set with a probability of at least $1 - \delta$ using $8\alpha d_{\max}\left(KA|\mathsf{An}(Y)| + B\right) \log\left(|\mathsf{An}(Y)|\right)$ interventions, where $A$ and $B$ are given by line 2 of Algorithm 1, and $C$ is given by line 3 of Algorithm 3 by setting $\alpha = \frac{2d_{\max} \log(\frac{4}{\delta} + 2)}{\log|\mathsf{An}(Y)|}$, $\delta_1 = \frac{\delta}{64\alpha d_{\max}|\mathsf{An}(Y)| \log|\mathsf{An}(Y)|}$, and $\delta_2 = \delta_3 = \delta_4 = \frac{\delta}{64\alpha d_{\max} \log|\mathsf{An}(Y)|}$.

The last phase is just running the UCB algorithm over the set of all possibly optimal arms, i.e., $\mathcal{A} = \{\Omega(I) \mid \forall I \in \mathcal{I}_{\mathcal{G}}\}$. This phase has a regret bound of $\sum_{\mathbf{s} \in \{\Omega(I) \mid \forall I \in \mathcal{I}_{\mathcal{G}}\}} \Delta_{do(\mathbf{s})} \left(1 + \frac{\log T}{\Delta_{do(\mathbf{s})}^2}\right)$ [26]. Now combining all the results we have the following:

Algorithm 4 learns the true set of POMISs $\mathcal{I}_{\mathcal{G}}$ with probability at least $1 - \delta - \delta = 1 - 2\delta$, and under the good event $E$ that it learns POMISs correctly, the cumulative regret is bounded as follows:

$$R_t \leq Kn \max\left(\frac{8}{\epsilon^2}, \frac{8}{\gamma^2}\right) \log \frac{4n^2 K^2}{\delta} + \frac{8}{\epsilon^2} \log \frac{4nK^2}{\delta}$$

(59)

$$+ 8\alpha d_{max}\left(KA|\mathsf{An}(Y)| + \max(B, C)\right) \log\left(|\mathsf{An}(Y)|\right) + \sum_{\mathbf{s} \in \{\Omega(I) \mid \forall I \in \mathcal{I}_{\mathcal{G}}\}} \Delta_{do(\mathbf{s})} \left(1 + \frac{\log T}{\Delta_{do(\mathbf{s})}^2}\right),$$

where $A$ and $B$ are given by line 2 of Algorithm 1, and $C$ is given by line 3 of Algorithm 3 by setting $\alpha = \frac{2d_{max} \log\left(\frac{4}{\delta}+2\right)}{\log|\mathsf{An}(Y)|}$, $\delta_1 = \frac{\delta}{64\alpha d_{max}|\mathsf{An}(Y)| \log|\mathsf{An}(Y)|}$ and $\delta_2 = \delta_3 = \delta_4 = \frac{\delta}{64\alpha d_{max} \log|\mathsf{An}(Y)|}$. This completes the proof of the Theorem 5.1.

### A.14 Comparison with SCM-based Approximate Allocation Matching Algorithm from [6]:

Our proposed algorithm, Algorithm 4, consists of two phases. The first phase uses interventional samples to learn the set of POMISs, and the second phase uses the UCB algorithm to find the optimal arm among the POMISs. Note that in the second phase, we use the UCB algorithm, which assumes that arms are independent of one another. However, in the case of causal bandits, the arms are correlated, and every intervention provides some information about other interventions. The UCB algorithm cannot exploit this information. However, [6] proposes an algorithm to exploit the correlations between arms in a causal bandit setting, which accelerates the learning compared to the simple UCB algorithm. The main limitation is that the algorithm requires access to the true causal graph. Therefore, it is possible that we can use an alternative approach where instead of POMISs, we learn the entire causal graph and then use the SCM-based Approximate Allocation Matching Algorithm from [6] for our problem setup. This approach can also allow us to reuse the intervention samples from the discovery phase to accelerate the next phase. However, the main drawback of this approach is that the algorithm proposed in [6] faces issues when it comes to larger, densely connected causal graphs. We explain the reasoning of our claim by reviewing some concepts from the paper [6].

In order to exploit the correlations between different arms in a causal bandit setting, the authors in [6] rely on response variable formulation for causal effects, which we discuss very briefly here. For any causal graph $\mathcal{G}$, the observed variables $\mathbf{V}$ can be uniquely partitioned into c-components $\mathbf{C}^1, \ldots, \mathbf{C}^{n_c(\mathcal{G})}$. Consider a set of response variables $\mathbf{M}$, which we also partition into $\mathbf{M}^1, \ldots, \mathbf{M}^{n_c(\mathcal{G})}$, where each $\mathbf{M}^j$ contains response variables corresponding to every observed variable in the corresponding c-component $\mathbf{C}^j$. Within a c-component, the response variables of all the observed variables are correlated since they are connected by bidirected edges. However, across two c-components, the response variables are independent. As a result, $P(\mathbf{m}) = \prod_{j=1}^{n_c(\mathcal{G})} P(\mathbf{m}^j)$. By concatenating $P(\mathbf{m}^j)$ for each $\mathbf{m}^j \in \Omega(\mathbf{M}^j)$, one can construct a vector $\mathbf{p}_j \in \Delta(|\Omega(\mathbf{M}^j)|)$ where $\Delta(|\Omega(\mathbf{M}^j)|)$ denotes the probability simplex over the discrete domain $\Omega(\mathbf{M}^j)$. Let the parent set of a c-component $\mathbf{C}^j$ be $\mathsf{Pa}_{\mathbf{C}^j} := \left(\bigcup_{i:V_i \in \mathbf{C}^j} \mathsf{Pa}_i\right) \setminus \mathbf{C}^j$. When taking intervention $do(\mathbf{S} = \mathbf{s})$, the values of $\mathbf{C}^j \cap \mathbf{S}$ are set to $\mathbf{s}[\mathbf{C}^j]$, which denotes the values of $\mathbf{C}^j \cap \mathbf{S}$ that are consistent with $\mathbf{s}$. $\mathbf{M}^j$ picks the mapping functions from $\mathsf{Pa}_i$ to $V_i$ for all $V_i \in \mathbf{C}^j$. By marking configurations in $B_{\mathcal{G}, \mathbf{s}[\mathbf{C}^j]}(\mathbf{C}^j, \mathsf{pa}_{\mathbf{C}^j}) \subseteq \Omega(\mathbf{M}^j)$ with 1 and 0, one constructs a vector $b_{\mathcal{G}, \mathbf{s}[\mathbf{C}^j]}(\mathbf{C}^j, \mathsf{pa}_{\mathbf{C}^j}) \in \{0, 1\}^{|\Omega(\mathbf{M}^j)|}$ such that:

$$P_{\mathbf{s}}(\mathbf{v}) = \prod_{j=1}^{n_c(\mathcal{G})} P\left(\mathbf{M}^j \in B_{\mathcal{G}, \mathbf{s}[\mathbf{C}^j]}(\mathbf{C}^j, \mathsf{pa}_{\mathbf{C}^j})\right) = \prod_{j=1}^{n_c(\mathcal{G})} b_{\mathcal{G}, \mathbf{s}[\mathbf{C}^j]}^{\top}(\mathbf{C}^j, \mathsf{pa}_{\mathbf{C}^j}) \mathbf{p}_j. \quad (60)$$

The equation 60 is very useful since it enables us to exploit the correlations between different interventions in the causal bandit setting. This is because every interventional distribution can be

written as a deterministic linear function of the response variable distribution. Thus, it is possible that using the response variable decomposition, we can reuse the samples from discovery into the next phase and accelerate learning of the optimal arm. However, we need to discuss the scalability of this approach. Note that every variable in the SCM can take values from the set $[K]$, and in total, there could be $K$ different realizations for $V_j$ for every realization of its parents $\mathsf{Pa}(V_j)$. As a result, there are a total of $K^{K^{|\mathsf{Pa}(V_j)|}}$ possible mappings from $\mathsf{Pa}(V_j)$ to $V_j$. Also, note that within a c-component, the response variables for every observed variable are correlated. This implies that for every component $\mathbf{C}^j$, the corresponding response variable $\mathbf{M}^j$ has the domain $|\Omega(\mathbf{M}^j)| = \prod_{V_i \in \mathbf{C}^j} K^{K^{|\mathsf{Pa}(V_i)|}}$. Thus, every vector $\mathbf{p}_j$ will have a total of $\prod_{V_i \in \mathbf{C}^j} K^{K^{|\mathsf{Pa}(V_i)|}}$ components. Although the response variable decomposition is useful for smaller and sparse causal graphs, the scaling for the length of vectors $\mathbf{p}_j$ is clearly exponential, making the use of response variable decomposition infeasible for larger or denser causal graphs. All in all, there are correlations between different arms in causal bandits, but it is not clear how to exploit them effectively, especially for larger and denser causal graphs, which is still an open problem.

### A.15   Experimental Compute Resources and Runtime

We ran our experiments on a server equipped with the AMD Ryzen Threadripper PRO 5995WX CPU, which has 64 cores and 128 threads, with a base clock speed of 2.7 GHz and a maximum boost clock speed of 4.5 GHz, along with 128 GB of RAM. The total runtime for the experimental plots in Figures 2 and 3 is around 2 hours. For the experimental plots in Figure 4, the total runtime is around 2 hours for each subplot since we run the full algorithm for multiple randomly sampled graphs.

### A.16   Broader Impacts of our Work

This paper presents work with the goal of advancing the field of Machine Learning. Since the causal bandit framework can be used to model real-life decision-making scenarios, there are some potential societal consequences of our work. The possibility of biased or incomplete understanding of causal relationships could lead to misguided decision-making or policy recommendations in real-world situations. Thus, extra care and consideration of ethical boundaries regarding actions/interventions are needed while applying our proposed methodology to practical problems.

