# OpenReview forum: "Partial Structure Discovery is Sufficient for No-regret Learning in Causal Bandits"
_NeurIPS.cc/2024/Conference — NeurIPS 2024 poster_

### Official Review · Reviewer_FQKc · 2024-07-10

**Soundness:** 4
**Presentation:** 3
**Contribution:** 3
**Rating:** 7
**Confidence:** 3

**Summary:**

The paper addresses causal bandits problem without prior knowledge of causal graph and in the presence of latent variables. Previous work shows that the best possible interventions form a set known as POMISs. The paper theoretically demonstrates that a partial discovery of the causal graph is sufficient to find POMISs.
To tackle the main problem, the authors propose a randomized algorithm to identify POMISs without complete discovery and then use the UCB algorithm on the candidate set.

**Strengths:**

1- The paper considers an important problem in the causal bandits area, in which the graph is unknown and latent variables exist.

2- To find optimal candidate interventions (POMISs) the paper demonstrated that a partial discovery of the main causal graph is sufficient. (Theorem 3.1)

3- A two-phased algorithm has been proposed for solving the causal bandit by providing a precise theoretical analysis.

**Weaknesses:**

1- The proposed regret bound can become very large when either $\epsilon$ or $\gamma$ is small, which is a possible scenario.

2- It’s not clear whether the proposed algorithm and upper is optimal or not. No discussion has been provided.

3- There is not any discussion regarding the running time of algorithms.

**Questions:**

Could you discuss the points mentioned above?

1- The algorithms (e.g., Algorithm 4.1) are using the values in assumptions 4.1 and 4.2. How is it possible to access this information?

2- How do you compare your algorithm (the regret bound) with a naive algorithm that blindly iterates over all possible interventions? Is it possible the latter performs better than your algorithm? If yes, which types of graphs do have this property? (I read Section A.15)

3- In algorithm 2, for each iteration, Should not set $W = \emptyset$?

**Limitations:**

Yes.

---

> ### Author Rebuttal · Authors · 2024-08-07
>
> We would like to thank the reviewer for their detailed review and comments. Below we address the reviewer’s concerns:
>
> **Proposed regret bound can become very large when  $\epsilon$  , $\gamma$  is small. How is it possible to know  $\epsilon$
>  , $\gamma$ in-advance?**
>
> We agree with the reviewer that the regret bound depends on $\epsilon$ and $\gamma$. However, in the context of causal bandits, the main emphasis is ensuring that regret is sublinear in rounds and does not grow exponentially with the size of the problem instance, which is the number of nodes $n$ in the graph. Also, $\epsilon$ and $\gamma$ are constants which do not have any dependence on the number of rounds or nodes in the graph. Such gaps parameters are fundamental, as there is no way around them. When these gaps are small, any statistical independent test would require a large number of samples for reliable conclusions.
>
> In the introduction section, we discuss that the majority of existing work on causal bandits assumes the causal graph is known, which can be unrealistic in many practical applications. This is due to the challenges associated with sample-efficient causal discovery with theoretical guarantees. The previous two papers we discuss, which deal with causal bandits with an unknown graph, are Lu et al. (2021) (cited as [7]) and Konobeev et al. (2023) (cited as [9]). Both papers also have gap assumptions, such as assumption 2 and assumption 3 in Lu et al. (2021), and assumption 4.1 and 4.2 in Konobeev et al. The novelty of our work is that both papers assume causal sufficiency and do not allow for the presence of confounders. Additionally, the regret bounds in both papers have similar scaling with respect to gap parameters.
>
> Our algorithm indeed requires access to the gap parameters $\epsilon$ and $\gamma$, similar to the approaches in [7] and [9]. If these parameters are unknown in practice, one possible solution is to start with some suitable initial values for the gap parameters. Then, you can alternate between the learning phase and the UCB phase, gradually adjusting the values of the gap parameters until a required performance criterion is met.
>
> **On optimality of proposed algorithm and regret bound:**
>
> There is prior work on lower bounds and optimality analysis for causal bandits with known graphs. However, **with unknown graphs**—especially when **latent confounders** are present—**lower bounds or optimality analysis** remains an **open problem**. This is because causal discovery is required as part of the solution; without graph learning, one must consider interventions on all possible subsets of nodes. Additionally, finite sample discovery for causal graphs with confounders, along with theoretical guarantees, is still an open problem. Our randomized algorithm makes progress toward addressing this under mild assumptions. Overall, optimal analysis for the learning phase of our proposed algorithm and its upper bound remains a topic for future work.
>
> **On Run time of algorithms:**
>
> We provide details about computational resources and runtime for our simulations in supplementary material, section A.17. Asymptotically, our full bandit algorithm has polynomial runtime complexity, i.e., $O(\text{poly}(n))$, where $n$ represents the number of nodes in the causal bandits problem. However, please note that the runtime reported in A.17 is larger because it accounts for the total time across multiple randomly sampled graphs with varying numbers of nodes. We will include discussion on the runtime complexity of our algorithm in revised manuscript.
>
>
> **Algorithm 2, for each iteration, Should not set $\mathbf{W} = \phi$?**
>
> There is a typo on line 4; it should be $\mathbf{W} = \phi$. We set $\mathbf{W} = \phi$ for every iteration of the outer loop, and then the target set $\mathbf{W}$ is randomly sampled for each iteration of the outer loop. At every iteration, Algorithm 2 calls Algorithm 1 to compute the transitive closure of the post-intervention graph $\mathcal{G}^{tc}_{\overline{\mathbf{W}}}$ and finally accumulates all edges found in the transitive reduction of post-interventional graphs across all iterations of the outer loop.
>
> **On comparison with  naive algorithm that blindly iterates over all possible interventions? Can it outperform our Algorithm?**
>
> The key feature of our regret bound is that it is sublinear in the number of rounds and has polynomial scaling with respect to the number of nodes $n$ in the graph. In comparison to a naive algorithm that intervenes on all possible subsets of nodes using any bandit algorithm, we at least have to play each arm once. For instance, if the causal graph has $n$ nodes, there will be
> $ \sum_{i=1}^{n} \binom{n}{i} K^i = (K+1)^n
> $ different possible arms/interventions, where $K$ is the number of different possible values each node can take. The advantage of sample-efficient discovery is that it helps us reduce the action space before applying the UCB algorithm. We have already included this discussion in the paper on lines 311 to 313.
>
> In our simulations, we show that once the number of nodes exceeds a certain number, around 17 or so, the number of arms for the naive algorithm grows much larger than our regret bound, as shown in the plot in Fig. 3. This is because the number of arms for a naive algorithm scales exponentially with the number of nodes $n$, whereas our regret bound has polynomial scaling with respect to $n$. The naive algorithm may perform better than our solution for graphs with a small number of nodes; however, as the number of nodes grows, it will not even be feasible to run the naive algorithm that does not consider causal information because of the exponential size of the action space.
>
>
> We hope our response has clarified the reviewer's concerns and respectfully request that they reconsider their evaluation in light of our response. We would be more than pleased to engage in further discussion if the reviewer has any additional concerns or questions.

---

> > ### Comment · Reviewer_FQKc · 2024-08-11
> >
> > Thank you for your response. Regarding the comparison with Classic UCB, I understand that your algorithm reduces the action space before applying UCB. However, this typically incurs a cost (as reflected in the first four terms of your regret bound). My question was: under which graph structures and parametric assumptions (i.e., $\epsilon$ and  $\gamma$) would it be preferable to skip causal discovery and apply only classic UCB? Could you discuss this?

---

> ### Author Response · Authors · 2024-08-11
> **Re.**
>
> Thank you for your reply. The **main deciding factor** in the choice between vanilla UCB and our proposed algorithm is the number of nodes, $ n $, in the graph. The vanilla UCB may perform better for graphs with a small number of nodes. However, as the number of nodes increases, it becomes infeasible to run the naive algorithm that does not consider causal information due to the exponential growth of the action space. Our simulations demonstrate that once the number of nodes exceeds a certain threshold—around 17 or so—the number of arms for the vanilla UCB algorithm grows significantly larger than our regret bound, as shown in the plot in Fig. 3. Note that we set the gap parameters $ \gamma = \epsilon = 0.01 $ in our simulations. The number of arms for the vanilla UCB algorithm scales exponentially with the number of nodes $ n $, whereas our regret bound scales polynomially with respect to $ n $.
>
> **Impact of  graph structure:**
>
> Regarding the graph structure, it depends on whether we have prior knowledge of the underlying graph structure before running the bandit algorithm. For instance, if we know that there is a confounder between every pair of nodes in the graph, causal discovery will not reduce the size of the action space, and it might be preferable to skip causal discovery and apply only the classic UCB. The key factor here is that the reduction of the action space using causal discovery depends on the density of latent confounders. If there are confounders between a large number of pairs of nodes, the reduction in the size of the action space becomes smaller. However, without prior knowledge of the graph's structure, one cannot make this distinction. We will include a simulation with a range of values for the parameter $\rho_L$, from $0.10$ to $1.00$, to demonstrate how the reduction in the size of the action space achieved through causal discovery—compared to vanilla UCB—varies with the density of latent confounders in the underlying graph. Note that the parameter $\rho_L$ controls the density of latent confounders in the sampled graph. We are grateful to the reviewer for highlighting this point and we believe it would be a valuable addition to our manuscript.
>
> **Impact of  parametric assumptions (i.e., $\epsilon$ and  $\gamma$):**
>
> Regarding the parametric gap assumptions, $\epsilon$ and $\gamma$, they are constants that do not depend on the number of nodes in the graphs. As discussed earlier, running UCB with an exponential number of arms, specifically $(K+1)^n$, will result in very large regret for large $n$. However, we can still make a basic comparison between vanilla UCB and our algorithm by comparing the sum of the first four terms in our regret bound with the regret bound or the number of arms in the vanilla UCB algorithm, similar to Figure 3 in our paper. This comparison will help inform our choice. We will include this discussion in the revised manuscript.
>
> We hope that we have addressed the reviewer’s question and would be happy to discuss further if there are any follow-up questions. If the reviewer has no further questions, we would kindly request them to reconsider our score, taking into account our rebuttal and the additions we have suggested.

---

> > ### Comment · Reviewer_FQKc · 2024-08-11
> >
> > Thank you for your comprehensive discussion. After reviewing the rebuttals and considering the comments from other reviewers, I intend to raise my score to 7.

---

> ### Author Response · Authors · 2024-08-11
> **Re.**
>
> We thank the reviewer for their insightful comments and suggestions to improve the clarity of our manuscript. We are also grateful to the reviewer for increasing our score.

---

### Official Review · Reviewer_VRJg · 2024-07-10

**Soundness:** 4
**Presentation:** 4
**Contribution:** 3
**Rating:** 6
**Confidence:** 3

**Summary:**

The authors extend the methodology in Lee and Bareinboim [2018] (L&B) for regret minimisation in a causal bandit setting with initially unknown causal structure and possible latent confounding. They identify the necessary and sufficient structural knowledge of latent confounding to identify possibly optimal minimal intervention sets (POMISs), given full knowledge of the observed causal graph, under similar modelling assumptions to L&B.

While the conditions do not improve the number of test interventions needed in worst-case scenario, they can offer super-exponential speedup in other scenarios. The authors provide a finite sample POMIS-discovery algorithm with coverage guarantees of the true set of POMISs and derive regret upper bounds. Experiments, particularly figure 2, show a clear (regret) advantage over learning all latent confounders in a setting with random, triangulated DAGs.

**Strengths:**

The authors present their contributions clearly, logically covering their motivation, theoretical results, algorithmic design, coverage guarantees, regret upper bounds, experimental design and experimental results. It is clear how their approach works in conjunction with other stages such as learning the observable graph and regret minimisation with the full set of POMISs.

Experiments convincingly show the advantage of partial structure discovery over testing for all latent confounding in a class of random DAGs.

**Weaknesses:**

(Latent structure assumption) A key modelling assumption, retained from L&B, is that latent variables are mutually independent and confound only a pair of measured variables each. This modelling assumption (why is it necessary? how restrictive is it mathematically? in what real-world setting could it be relevant?) was not sufficiently explored in L&B or by the authors. The assumption is especially relevant to the impact of the authors' contributions so should be discussed more thoroughly in the text.

**Questions:**

(Latent structure assumption) What is the role of the assumed structure on latent confounding in your results and how restrictive actually are these assumptions? Can these assumptions be relaxed to some degree, say if latent confounders are allowed to confound up to n (e.g., 3) observed variables, or have bounded pairwise correlation? Is it permitted for three vertices $(V_1, V_2, V_3)$ to be confounded pairwise, i.e., for confounding between $(V_1, V_2)$, $(V_2,V_3)$ and $(V_1, V_3)$?

Are the graphs generated for the simulation experiments - i.e., after triangulation - typically dense or sparse? Is this an important factor to understand the advantage of your method over learning all latent confounders? Perhaps a parameter sweep over $\rho$ and $\rho_L$, i.e., a property of the observed versus unobserved graphs, could be informative to this end. Was the triangulated structure of the observed graph important for your results?

**Limitations:**

Yes - the authors state all underlying assumptions of their model. I would like to understand whether the latent structure assumption is limiting for real-world applications.

---

> ### Author Rebuttal · Authors · 2024-08-07
>
> We would like to thank the reviewer for their detailed review and comments. Below we address the reviewer’s concerns:
>
> **(Latent structure assumption)  Role and how restrictive it is? Can these assumptions be relaxed , say if latent confounders are allowed to confound up to n (e.g., 3) observed variables, or have bounded pairwise correlation? Can three vertices   to be confounded pairwise?**
>
> The latent structure assumption we are using, also known as the semi-Markovian assumption, is widely used in the causality literature, including in L\&B (additional references are provided at the end). We will briefly explain how general this assumption is.
>
> An SCM (Structural Causal Model) consists of a set of equations of the form $ X_i = f_i(\text{Pa}(X_i), U_i) $, where $ U_i $ are the exogenous noise variables. If the noise variables are independent, the model is Markovian. However, when exogenous noise variables are dependent, we have a broader class of models which are not Markovian. These are called semi-Markovian causal models. For instance, if exogenous variables $ U_i $ and $ U_j $ are dependent, we say $ X_i $ and $ X_j $ are confounded. We assume the underlying SCM for the causal bandit problem is semi-Markovian, which entails a specific latent structure: the latent variables are root nodes with no parents and can have at most two observed variables as children. A latent variable with two observed children implies that the observed variables have a latent confounder between them.
>
>
> Coming to the second part of the question: The semi-Markovian assumption allows for pairwise confounding between any pair of observed variables. The main restriction is that one exogenous variable, for instance, one latent variable $ L $, cannot be the parent of three or more observed variables $ v_1, v_2, v_3, \ldots $. However, one can map this non-Markovian example to a Markovian model by assuming pairwise confounding between the nodes. Also, if one decides to run the algorithm, they would detect that these variables have pairwise confounders. The characterization of possibly optimal arms (POMISs) in the paper by L\&B is based on the assumption that the underlying causal model is semi-Markovian because they use existing identifiability results for semi-Markovian causal models. Our algorithm builds upon their results. A general extension of this work to non-Markovian models and the implications of violating the semi-Markovian assumption remain future work for us. Overall, we agree with the reviewer that it is essential to highlight this assumption and its implications in the revised manuscript.
>
>
> **[1]** *Pearl, J., 2009. Causality. Cambridge university press.*
>
> **[2]** *Tian, J. and Pearl, J., 2002, August. A general identification condition for causal effects. In Aaai/iaai (pp. 567-573).*
>
> **[3]** *Bareinboim, E., Correa, J.D., Ibeling, D. and Icard, T., 2022. On Pearl’s hierarchy and the foundations of causal inference. In Probabilistic and causal inference: the works of judea pearl (pp. 507-556).*
>
> **[4]** *Meganck, S., Maes, S., Leray, P., Manderick, B., Rouen, I.N.S.A. and St Etienne du Rouvray, F., 2006. Learning Semi-Markovian Causal Models using Experiments. In Probabilistic Graphical Models (pp. 195-206).*
>
>
> **Are the graphs generated after triangulation - typically dense or sparse? Is this an important factor to understand the advantage of over learning all latent confounders? Perhaps a parameter sweep over $\rho$ and $\rho_L$
>  could be informative to this end.**
>
> In the process of chordalization or triangulation, additional edges are added to make the observed graph structure chordal, which generally makes observable graphs a little denser. There are two density parameters: $\rho$, which controls the density of the observable graph, and $\rho_L$, which controls the density of latent confounders. Specifically, after sampling the observable graph, we add latents between pairs of observed variables with a probability of $\rho_L$. While the triangulation makes the observable graph somewhat denser, it does not play a major role in our simulation studies and is optional.
>
> The main idea we are trying to show through simulations is the advantage of partial discovery over full discovery for the causal bandit problem with an unknown graph.  We agree with the reviewer that a parameter sweep for different values of the density parameters $\rho$ and $\rho_L$ can be useful in our simulation section. In fact, we have shown the effect of a parameter sweep in our results in Section A.14 of the supplementary material. We set the variables $\rho$ and $\rho_L$ to values $0.2, 0.4, 0.6$ and plotted a grid with results for all 9 possible combinations. The main conclusion was that the advantage we get with partial discovery over full discovery mainly depends on the latent density parameter $\rho_L$, and it reduces as we increase $\rho_L$ from $0.2$ to $0.6$. The trend remains consistent across different values of the observable graph density parameter $\rho$. We did not include these plots in the main paper due to space constraints, but we will move them to the camera-ready version using the extra content page.
>
> We hope our response has clarified the reviewer's concerns and respectfully request that they reconsider their evaluation in light of our response. We would be more than pleased to engage in further discussion if the reviewer has any additional concerns or questions.

---

> > ### Comment · Reviewer_VRJg · 2024-08-12
> >
> > Many thanks for the clarification and extended experiments. I maintain my score for a weak accept.

---

> > > ### Author Response · Authors · 2024-08-12
> > > **Re.**
> > >
> > > We thank the reviewer for their insightful comments and suggestions. We will include the extended experiments in the main paper following the reviewer's suggestion. We are also grateful to the reviewer for recommending acceptance of our work.

---

### Official Review · Reviewer_MSdE · 2024-07-13

**Soundness:** 4
**Presentation:** 3
**Contribution:** 4
**Rating:** 7
**Confidence:** 3

**Summary:**

This paper ooks into the problem of causal bandits without graph information and with unobserved variables. The paper presents both graph learning algorithms and causal bandit algorithms.

**Strengths:**

This paper tackles the novel problem of causal bandits (CB) with unknown graphs and unobserved variables. Theorem 3.1 is particularly interesting, and the use of a randomized algorithm to sample-efficiently learn the causal graphs is notable.

**Weaknesses:**

There are a few typos in the paper:
- For the definition of transitive reduction, does it force $\mathcal{G}$ to be connected? Otherwise, the empty edge set is minimum.

- For the d-separation defined in line 119, shouldn't it condition on $\mathbf{W}$?

- Assumption 2.1 is defined on a mixture of hard intervention and stochastic intervention (in line 126)?

- Line 4 in Algorithm 2, should it be $\mathcal{W}=\emptyset$?

**Questions:**

- How is transitive reduction computed in Algorithm 2? Does it need to exhaust all possible sub-graphs?

- Correct me if I am wrong: In Assumption 4.1 and Assumption 4.2, the assumption is made for **any** $\mathbf{w}\in [K]^{\mathbf{W}}$, which is stronger than the other causal bandit papers, e.g. [7], [9].

- For Algorithm 3, line 4, how is the set $\mathbf{W}$ determined as it is neither an input nor the output of LearnObservableGraph? Te $\mathbf{W}$ matrices in $\mathcal{I}Data$ does not share the same $\mathbf{W}$, and and line 6 seems to attempt to find $\mathbf{W}$.

**Limitations:**

The paper assumes semi-Markovian causal models and stronger identification assumptions.

---

> ### Author Rebuttal · Authors · 2024-08-07
>
> We would like to thank the reviewer for their detailed review and comments. Below we address the reviewer’s concerns:
>
> **Transitive reduction forces $\mathcal{G}$  to be connected?**
>
> Transitive reduction of a DAG $\mathcal{G}$ is another DAG with the same transitive closure as  $\mathcal{G}$ but with the least number of edges. The transitive closure of $\mathcal{G}$ is another DAG $\mathcal{G}^{tc}$, where a directed edge $ V_i \rightarrow V_j $ is included in $\mathcal{G}^{tc}$ iff $ V_i \in \mathsf{An}(V_j) $ (as defined in line 112). The definition does not restrict $\mathcal{G}$ to be connected. The transitive reduction will be an empty edge set iff $\mathcal{G}$ itself has no edges, because the transitive reduction must have the same transitive closure as $\mathcal{G}$.
>
> **d-separation in line 119, shouldn't it condition on $\mathbf{W}$ ?**
> **Line 4 in Alg. 2, should be $\mathbf{W} = \phi$.**
>
> Apologies for the typos in line 119 and Algorithm 2; we will fix them
>
>
> **How is transitive reduction computed?**
>
>
> We use a polynomial time $ O(poly(n)) $ algorithm from  [1] . In Line 227 we cite this paper and recall the results from Theorem 1 and Theorem 3: The transitive reduction of a DAG is unique and can be computed in polynomial time. Specifically, Theorem 3  provides a polynomial-time algorithm to compute transitive reduction using polynomial-time algorithms to find the transitive closure, such as the Floyd-Warshall algorithm having  complexity of $ O(poly(n)) $. We will clarify this in Algorithm 2 as well.
>
>
> **[1]** *Alfred V. Aho, Michael R Garey, and Jeffrey D. Ullman. The transitive reduction of a directed
> graph. SIAM Journal on Computing, 1972.*
>
>
>
> **Algorithm 3, line 4, how is the set $\mathbf{W}$  determined, and line 6  attempts to find $\mathbf{W}$ in $\mathcal{I}Data$:**
>
>
> We apologize, there is a typo in line 4 of Algorithm 3. It should only be: (For all pairs $V_i, V_j \in \mathbf{V}$) without any mention of $\mathbf{W}$. Later on, we attempt to find interventional samples for some suitable $\mathbf{W}$ in $\mathcal{I}\text{Data}$ where $\mathbf{W}$ is the union of parent sets of $V_i$ and $V_j$ excluding $V_i$ itself. Note that in the proof of Theorem 2, we show that such target $\mathbf{W}$ for any pair of nodes will be selected in Algorithm 2 that is generating $\mathcal{I}\text{Data}$ with high probability (for details refer to Section A.12). We will fix the typo in revised manuscript and clarify it further in revised manuscript.
>
> **Assumption 2.1 is defined on a mixture of hard intervention and stochastic intervention?**
> **In Assumption 4.1 and Assumption 4.2, the assumption is made for any
> $\mathbf{w} \in [K]^{|\mathbf{W}|}$, which is stronger than [7], [9]?**
>
>
> We will address the reviewer’s concerns regarding Assumptions 2.1, 4.1, and 4.2 together. Assumption 2.1 is the main assumption in the paper. Using Assumption 2.1, we state and prove two lemmas, 4.1 and 4.2, which can be used to detect ancestral relationships and the existence of confounders, respectively. To provide theoretical guarantees on sampling complexity, the inequality conditions in the lemmas alone are not sufficient. Therefore, we introduce gaps $\epsilon$ and $\gamma$ in Assumptions 4.1 and 4.2, using the results from Lemmas 4.1 and 4.2.
>
>
> We thank the reviewer for highlighting the point that Assumption 2.1 uses a mixture of hard and stochastic interventions. This can, in fact, be simplified by using only stochastic interventions instead. Consider a set of nodes $\mathbf{W} \subset \mathbf{V}$ and the stochastic intervention $do(\mathbf{W}, \mathbf{U})$ on $\mathbf{W}$ and any set $\mathbf{U} \subseteq \mathbf{V} \setminus \mathbf{W}$. The conditional independence (CI) statement $(\mathbf{X} \perp \mathbf{Y} \mid \mathbf{Z}){\small{\mathcal{M}}\tiny{{\mathbf{W}, \mathbf{U}}}}$ holds in the induced model if and only if there is a corresponding $d$-separation statement in post-interventional graph $(\mathbf{X} \perp_{d} \mathbf{Y} \mid \mathbf{Z}){\small{\mathcal{G}}\tiny{{\overline{\mathbf{W}, \mathbf{U}}}}}$ , where $\mathbf{X}$, $\mathbf{Y}$, and $\mathbf{Z}$ are disjoint subsets of $\mathbf{V} \setminus \mathbf{W}$. The CI statements are with respect to the post-interventional joint probability distribution. Assumption 2.1 is similar to the post-interventional faithfulness assumption used in the causality literature [1-3]. The difference pointed out by the reviewer between our Lemmas and Assumptions 4.1 and 4.2 compared to those in [7] and [9] stems from the post-interventional faithfulness Assumption 2.1.
>
> In [7], the reward node has only one parent, and the action space consists of single-node interventions. On the other hand, the regret bound in [9] includes a term with a factor of $ K^{|\text{Pa}(Y)|} $, which implies that the number of samples needed during the learning phase of the algorithm might grow exponentially with the number of nodes if the reward node has a large number of parents. In contrast, our regret bound has polynomial scaling with respect to the number of nodes in the graph.
>
>
> **[1]** *Alain Hauser and Peter Bühlmann. Two optimal strategies for active learning of causal networks
> from interventional data. In Proceedings of Sixth European Workshop on Probabilistic Graphical
> Models, 2012*
>
> **[2]**  *Wang, T.Z., Wu, X.Z., Huang, S.J. and Zhou, Z.H., 2020, November. Cost-effectively identifying causal effects when only response variable is observable. In International Conference on Machine Learning (pp. 10060-10069). PMLR*.
>
> **[3]**  *Kocaoglu, M., Shanmugam, K. and Bareinboim, E., 2017. Experimental design for learning causal graphs with latent variables. Advances in Neural Information Processing Systems*
>
>
> We hope our response has clarified the reviewer's concerns and respectfully request that they reconsider their evaluation in light of our response. We would be more than pleased to engage in further discussion if the reviewer has any additional concerns or questions.

---

> > ### Comment · Reviewer_MSdE · 2024-08-12
> >
> > Thanks the the response and I also read the other insightful discussions. I believe it is a solid paper that contributes to the field and I'd like to raise the score from 6 to 7.

---

> > > ### Author Response · Authors · 2024-08-12
> > > **Re.**
> > >
> > > We thank the reviewer for their insightful comments and suggestions. We are also grateful to the reviewer for appreciating our work and increasing the score.

---

### Decision · Program_Chairs · 2024-09-25

**Decision:**

Accept (poster)

**Comment:**

The paper reports a study on causal bandits with unknown causal graphs and latent confounders. It is an extension of the prior work by Lee and Bareinboim [5], who only considered the case of known causal graphs. The reviewers show positive support to the paper, acknowledging the contribution and importance of studying the causal bandits when the causal graphs are unknown and when there are latent confounders. Overall, I agree with the reviewers on the positive contribution of the paper elevating the research to the unknown causal graph case, and in particular, the results in Section 3 and 4 are interesting. However, during the final stage when I gave a check on the paper, I did find an issue concerning a claim of the paper that the regret bound is polynomial in the graph size $n$. This leads to some further discussions among the reviewers. After the discussion, we decided that the paper is worth an acceptance as the poster, but the authors have to remove this incorrect claim and thoroughly revise the paper.

The following is the detailed description of the issue:

The last sentence of the abstract make the following claim:

"We also establish regret bound for our two-phase approach, which is sublinear in the number of rounds and has polynomial scaling with respect to the number of nodes in the graph."

The authors also repeated this claim about polynomial dependency of the regret bound on the graph size several times in the rebuttal, to argue that they are better than the naive use of UCB directly without the causal discovery part, since the naive use of UCB will have the number of arms in the scale of $O(K^n)$ where $K$ is the number of possible values each variable can take, and $n$ is the number of observable variables in the causal graph.

But what they did is to find all possibly optimal minimum intervention sets (POMISs), following the work of [5]. It reduces significantly the number of possible intervention to explore, but it is unclear that it would reduce this number to a polynomial in $n$. Let the size of largest POMIS set is $P$ and the number of POMIS set is $Q$, the total number of arms for the MAB problem is $O(Q K^P)$. This can also be seen from the regret bound in Theorem 5.1: the last term needs to enumerate through all POMIS set $I$ in $\mathcal{I}_{\mathcal{G}}$, and for each $I$, it needs to enumerate through all value combinations of $I$, namely $\Omega(I)$. So it is exponential in $P$, and $Q$ could also be exponential. Unless the authors can claim that $P$ is a constant or $\log n$ term, and $Q$ is a polynomial in $n$, I don't think the authors can claim that their regret bound is polynomial in $n$. In fact it certainly exists some cases in which the possible enumeration above is exponential. I do not see any discussion in the paper or in the rebuttal about this fact, and there is no technical discussions on when the above number could be polynomial.

Another hint is from Figure 3 and 6 of the paper. The figure shows that the enumeration of all possible interventions is clear bad. However, looking at the enumeration of all possible interventions within the POMIS set, it is still like a linear curve with a small slope. But given that the x-axis is in linear scale while the y-axis is in log scale, I think it still indicates that the total number of enumerations for POMISs is exponential in $n$.

After some discussion with the reviewers, we believe that the claim that the regret bound has polynomial scaling to the graph size is incorrect and the authors should remove this claim from the paper. But the other contributions of the paper is strong enough to warrant the acceptance of the paper as a poster in NeurIPS.